# Estimating excess bound water content due to serpentinisation in mature slow-spreading oceanic crust using Vp/Vs

Lianjun Li [1] ✉, Jenny Collier [1] ✉, Tim Henstock [2] & Saskia Goes [1]

Mature oceanic crust carries chemically bound water which may be released in subduction zones or delivered to the deep mantle. Estimating water content in slow-spreading crust is challenging due to its complex lithology, requiring both P- and S-wave seismic velocity (Vp and Vs), the latter of which has been limited. Here we show 2D high-resolution Vp, Vs and excess bound water models due to serpentinisation of mature Atlantic crust near the Lesser Antilles. The ridge-parallel line crosses eight seafloor-spreading segments with equal numbers of magma-robust and magma-poor. Hydration is highly variable and mainly accommodated in strongly serpentinised peridotites, dominantly in magma-poor segments, which are not preferentially located near fracture zones. Serpentinised peridotites (17% of the crust) host four times more water than normal magmatic crust, increasing Atlantic subduction bound water budget by ~ 50%. This has implications back in geological time such as during supercontinent breakups when slow-spreading crust subduction was more common.

The subduction of hydrated oceanic lithosphere is a key driver of the solid Earth water cycle, transferring water from Earth's surface to the deep interior[1]. Water in the oceanic lithosphere can be stored in pores and cracks (free) or in hydrous minerals (chemically bound) within mafic crustal rocks, resulting from hydrothermal alteration[2,3]. Further hydration of the ultramafic upper mantle occurs during plate bending at the outer rise by the hydration of olivine to serpentine[4,5]. Any free water (particularly from the sediment component) is thought to be largely expelled at shallow depths, into the accretionary complex and overriding-plate crust[6,7]. The chemically bound water is released by metamorphic dehydration over a much larger depth range into the upper plate and mantle. The release of fluids from hydrous minerals triggers melting that fuels volcanism and affects the style of interplate coupling and the location and size of inter- and intraplate earthquakes[8]. Some of the bound water, especially that stored in serpentine, is stable to large depths in cold subducting slabs[9], allowing fluids to be carried beyond-arc depths[3,6]. Consequently, understanding the abundance and distribution of bound water in the incoming plate is important for estimates of the solid Earth water budgets.

Current understanding of water budgets is based on typical oceanic lithosphere, such as that which floors most of the Pacific, with mafic crust in a Penrose stratigraphy, namely an upper crust of basalt/dolerite with a high seismic velocity-depth gradient (-1 to 2/s, seismic layer 2) and a lower crust of gabbro with a low seismic velocity-depth gradient (-0.1 to 0.2/s, seismic layer 3)[10–12]. It has been estimated that such mafic crust can host about 2 wt% bound water in the upper and about 0.5 wt% in the lower crust[7,13,14]. The bound water content increases with age due to ongoing hydrothermal circulation and mineral precipitation within pores and cracks[7], and is thought to increase within fracture zones due to additional faulting at the ridge-transform axis[15]. This model of hydration, hereafter referred to as the 'Penrose hydration model', dominates calculations of subduction fluxes and models of slab dehydration[6,7,16,17]. While the Penrose hydration model is appropriate for the fast-spread lithosphere in the Pacific, in the Atlantic, slow-spreading ridges produce a more diverse crustal lithology as they alternate (in space and time) between magma-robust (magmatic) and magma-poor (tectonic) accretion modes. During magma-poor accretion, mantle (ultramafic) rocks can be

[1]Department of Earth Science and Engineering, Imperial College London, SW7 2BP, London, UK. [2]School of Ocean and Earth Science, University of Southampton, SO14 3ZH, Southampton, UK. ✉e-mail: l.li21@imperial.ac.uk; jenny.collier@imperial.ac.uk

exhumed along detachment faults[18–21] and form oceanic core complexes (OCCs). Once exposed to seawater, the serpentinisation of olivine within the ultramafics can bind up to 13 wt% water, about three times more than what can be stored in altered mafics[7,22,23]. Thus, OCCs, which are estimated to occupy about a quarter of the slow-spreading seafloor and more common at segment ends[21,24], could provide an important contribution to the subducting-plate water budget[24–27]. However, how much excess bound water, beyond the Penrose hydration model, is delivered from these OCCs has not yet been estimated. Whilst slow-spreading lithosphere is not dominant in modern-day subduction systems, it was likely more prevalent in the past and hence is an important endmember for understanding the solid Earth water cycle[25].

A key challenge in determining the hydration state of slow-spreading oceanic lithosphere, with its heterogeneous crustal composition, is that experimentally determined P-wave velocities (Vp) of mafic and hydrated ultramafic rocks overlap[28,29]. However, if the ratio of P- to S-wave seismic velocities (Vp/Vs) can be measured, it significantly improves our ability to discriminate between the degree of hydration and lithological variations[30] and other factors that may reduce Vp, such as fractures[31]. However, high-quality active-source marine S-wave studies are uncommon, as typically S-waves are recorded on just 20 to 30% of ocean bottom seismometers (OBS)[32] because there needs to be a suitable boundary to convert the source-generated P-waves. Such a boundary needs to have a high impedance contrast[33], e.g. the seafloor, the sediment-basement interface and/or the seismic layer 2A/2B interface in the young crust[34,35]. For mature crust, the

sediment-basement interface is generally the most efficient conversion interface as the Vp above the boundary is close to the Vs below[32,34,36,37]. Phases from other boundaries are either low-amplitude or obscured under the long 'ringing' coda of the S-waves converted from the sediment-basement interface[36]. The second requirement to record S-waves is either good coupling of the geophones with the seafloor and/or good conversion back from S- to P-waves so that they are present on hydrophone components. In this study, we acquired wide-angle data in a region with thick sediments, which is favourable for both phase conversion at the basement and geophone coupling. The data were collected with a dense receiver spacing, and ~85% of the OBS returned high-quality S-wave data suitable for travel time tomography.

Our study region lies in the western central Atlantic over crust that is 60–75 Ma old and was formed with an average half-spreading rate of ~16 mm/yr (Fig. 1)[38,39]. The seismic line (VoiLA line 2/3) was acquired as part of the VoiLA Project (Volatile Recycling in the Lesser Antilles)[40]. It is oriented approximately north-south and parallel to the isochrons. The line spans 384 km, from ~50 km north of the Fifteen-Twenty Fracture Zone (1520FZ) to ~60 km south of Marathon Fracture Zone (MFZ), crossing the Barracuda Trough, Barracuda Ridge, Tiburon Basin and Demerara Abyssal Plain (Fig. 1a). The Barracuda Ridge has been proposed to be originated as a transverse ridge flanking 1520FZ, similar to those along Vema, St. Paul and Romanche[41] formed by vertical tectonism[42] and it has been further deformed as a result of past compression across the North-American-South American plate boundary rather than volcanic construction[43,44]. The seismic line is

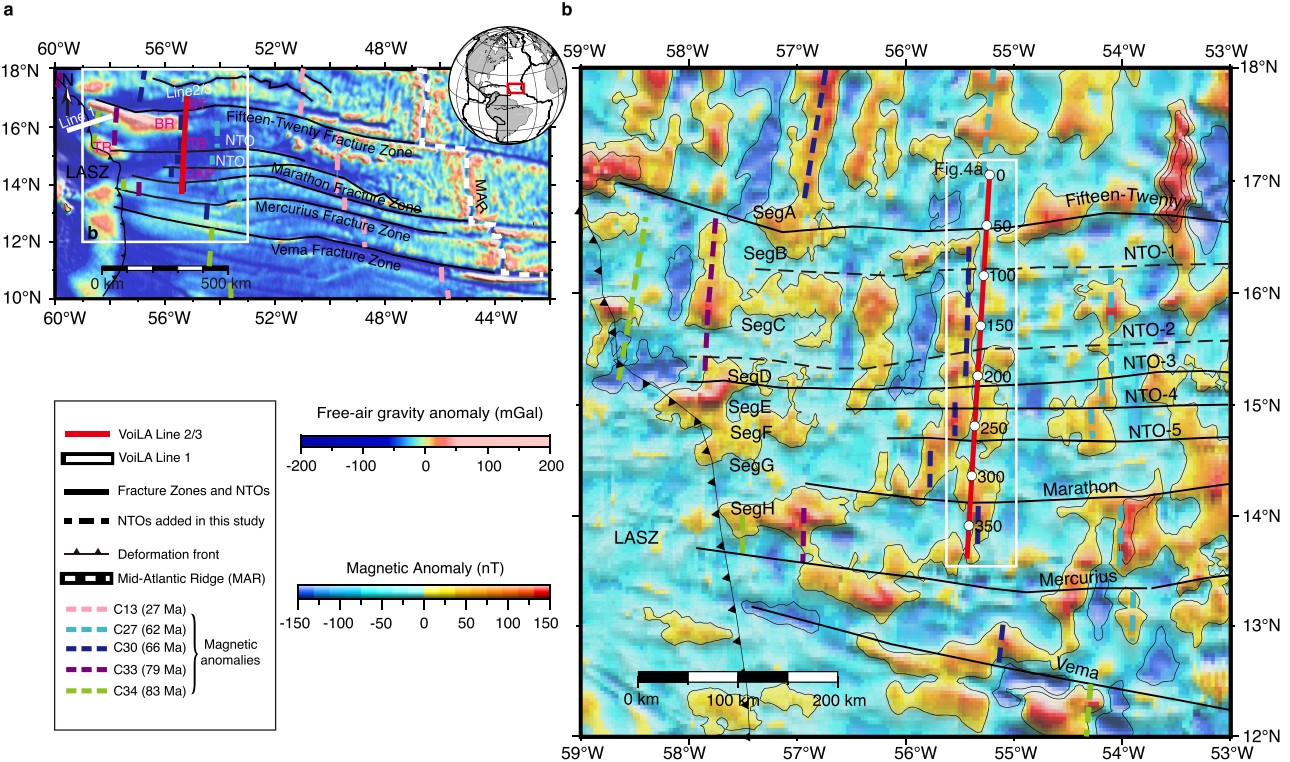

**Fig. 1 | Study region. a** Free-air gravity anomaly using data derived from satellite altimetry[47]. The red rectangle on the globe is the region shown in (**a**). Black lines are fracture zones (FZs) and non-transform discontinuities (NTOs) from ref. 88 and ref. 46 west of the Mid-Atlantic Ridge (MAR). Coloured dashed lines are seafloor spreading (magnetic) anomalies after ref. 46. The white solid line is the VoiLA Line 1 published in ref. 45. The red solid line is the VoiLA Line 2/3 used in this study. The white rectangle is the region shown in (**b**). VoiLA—Volatile Recycling in the Lesser Antilles[40]. **b** Magnetic anomaly grid from ref. 48 with thin lines in the background showing ship tracks used. Two dashed black lines are NTOs identified in this study, others were detailed by ref. 46. From north to south, seafloor spreading segments are named A to H, as indicated. Unless otherwise specified, the segmentations of the ridge-spreading fabrics established here are used in all other figures. The numbers labelled next to the solid white circles are model distances. The white rectangle is the region shown in Fig. 4a. LASZ Lesser Antilles subduction zone, BT Barracuda trough, BR Barracuda ridge, TR Tiburon rise, TB Tiburon basin, DAP Demerara Abyssal plain, Seg ridge-spreading segments.

~250 km east of the deformation front of the Lesser Antilles subduction zone (LASZ), ideal for assessing the hydration of incoming mature Atlantic oceanic lithosphere before bending into the subduction zone. Bending-related hydration was studied using data from VoiLA Line 1 (Fig. 1a)[45].

The seafloor spreading fabric in the region was previously assessed by ref. 46 using satellite altimetry-derived gravity (Fig. 1a)[47] and shipboard magnetics (Fig. 1b)[48]. By following their strategy, we interpreted two more second-order (non-transform offset, NTO) traces between the 1520FZ and NTO-3 (Fig. 1b). In total, the identified discontinuities bound eight seafloor-spreading segments, referred to as A to H from north to south, respectively. Unless otherwise specified, the segmentations of the ridge-spreading fabrics established here by potential field data are used in all further calculations and interpretations.

Davy et al.[46] presented a Vp model for Line 2/3 south of 15.5°N (our segments D-H) and found two diagnostic velocity-depth structures representing magma-robust and magma-poor controlled crustal accretion modes. The magmatic crust has a typical two-layer velocity-gradient structure, i.e. a Penrose model, while the tectonic crust is characterised by a single velocity-gradient structure with occasional velocity inversions identified as OCCs superimposed. Crustal thickness, as defined by a widespread PmP (Moho) reflector, was 2 to 3 km less in the magma-poor segments. On VoiLA Line 1, the duration of magma-robust and magma-poor episodes at the time of crustal accretion was estimated to be on the order of 3–5 Myr[45].

Here we present 2D high-resolution Vp and Vs models of mature Atlantic slow-spreading crust on the incoming plate of the LASZ. We first generate a Vp model and use it as a prior constraint for the Vs modelling (see 'Methods' section for details). Finally, we compare the Vp/Vs and Vp results to laboratory measurements of drilled samples and to theoretical fracture-water content templates. Together with constraints from gravity modelling, we identify OCCs in the mature crust and quantify the corresponding bound water content.

## Results

### Magma-robust and magma-poor segments

The final inverted Vp and Vs models are shown in Fig. 2a, b. The travel time tomography procedure followed to generate these models is detailed in the 'Methods' section and Supplementary Figs. 1–16.

The Vp and Vs models display strong lateral heterogeneities. Four regions with Vp inversions/reductions with depth are seen (at model distances 67, 140, 217 and 240 km, labelled as VI-1 to 4 respectively). These are considered candidates for OCCs where lower crustal and upper mantle materials are displaced to shallow depths. The most homogeneous velocity structure is found below the Demerara Abyssal Plain (south of model distance 280 km) and the Barracuda Trough (north of model distance 50 km). The average velocity and vertical velocity gradients are systematically higher within a given depth range in these two regions than in the rest of the line. Beneath the resolved Moho reflector, Vp and Vs is up to 7% lower below the Barracuda Ridge and northern Tiburon Basin (model distances 100 to 180 km) compared to elsewhere, where both Vp and Vs increase to normal mantle velocity within ~1 km below the Moho reflector.

Given the complexity of the velocity models, we applied a K-means cluster analysis for P- and S-wave velocity-depth attributes to aid the interpretation (see 'Methods' section and Supplementary Figs. 17–23). We found that the velocity-depth profiles can be grouped into four clusters with exclusive characters (shown in different colours in Fig. 3). The cluster in red (Fig. 3b, c) has a prominent two-layer seismic structure representing Layer 2 (the upper crust) and Layer 3 (the lower crust). It is the most dominant along the line, especially within Segments A, G and H (Fig. 3a). We termed it two-layer, and it has the least V-Z variations of all the clusters. The cluster in cyan (Fig. 3b) has prominent velocity inversions within ~2 km below the top basement and

relatively high velocity at a given depth compared to other clusters. It is termed Velocity-inversion and includes the four areas VI-1 to VI-4 identified previously, and has the most V-Z variability. The cluster in blue (Fig. 3b) consistently has the lowest velocity and a rather linear velocity gradient over the whole crustal column. It is termed Linear-gradient with a high presence in Segments B, C and F. The cluster in black (Fig. 3b, c) has a character intermediate between the two-layer and linear-gradient; therefore, it is termed Median-gradient. It is the most dominant cluster at segment ends and forms nearly 50% of Segments C and D. At NTOs, velocity-depth profiles are dominantly characterised by linear- and median-gradient clusters. At 1520FZ and MFZ, a mix of median and two-layer clusters is present, suggesting relatively higher velocity-depth gradients compared to those of NTOs. Note that although the cluster analysis was applied independently on Vp and Vs the results were broadly consistent with only minor discrepancies (Fig. 3a and Supplementary Fig. 24). Overall, we note there is not a strong contrast between segment ends and segment centres, but rather the variation seems to be on a segment-by-segment basis.

Having assigned clusters, the next step was to categorise the seafloor spreading mode based on the frequency of occurrence of each cluster within each segment (Supplementary Fig. 25). Segments A, G and H are considered magma-robust (magmatically accreted) because of the dominance of two-layer structure (Fig. 3c and Supplementary Figs. 25, 26a, f). Segments B, C, E and F are classified as magma-poor (tectonically accreted), characterised mainly by linear- and median-gradient clusters, and more importantly, velocity-inversion clusters (Fig. 3b and Supplementary Figs. 25, 26b, d, e). Note that in the previous visual classification[46], segment D was classified as magma-poor. Here, we interpret it as a magma-robust segment because only two-layer and median-gradient clusters are found within it.

In general, the average crustal thickness in the region is about 5.2 km, and the crust of magma-robust segments is consistently thicker than that of magma-poor segments (Fig. 3a). Within magma-robust segments, the P Moho reflector is interpreted as a petrological boundary between the crust and mantle[46]. Segments G and H have the thickest crust of ~6.4 km along the line. Segment D is exceptional with a thin crustal thickness of 4.4 km, similar to the surrounding magma-poor segments. The thin crust of this short segment could be due to its formation being at the beginning or end of a magma-robust supply cycle. Within magma-poor segments, crustal thickness is on average of ~4.4 km, estimated from only intermittently resolved Moho reflector. This is interpreted as an alteration Moho, which might be associated with the hydrothermal alteration of mantle peridotites, mafics or both[46]. Segment C has the thinnest crust with a thickness of only ~4.0 km, but PmP is hardly observed below about half of this segment. The Moho reflector resolved from SmS phases is generally ~0.7 to 1 km deeper than that from PmP phases (Fig. 3a) although they are present at similar offset ranges (Supplementary Fig. 6). Such a difference might be related to the influence of the properties and lithologies of the crust and/or Moho transition zone on P- and S-wave propagation. The depth uncertainty for the P Moho reflector could introduce an uncertainty of the crustal thickness of up to ±0.5 km (Supplementary Fig. 16a).

### Recognition of oceanic core complexes

With the benefit of Vp/Vs ratios, we can now re-examine the interpretation of potential oceanic core complexes (OCCs) made from Vp alone (VI-1-to-4). The velocity reduction could be most reasonably explained by serpentinisation[28,29] and/or increased porosity due to water-filled fractures[31,49]. To help discriminate between these two possibilities, we used gravity modelling as an independent diagnostic, given that the serpentinisation of ultramafics significantly reduces bulk density[50,51]. In contrast, although fractures can significantly reduce seismic properties, the total volume of fractures

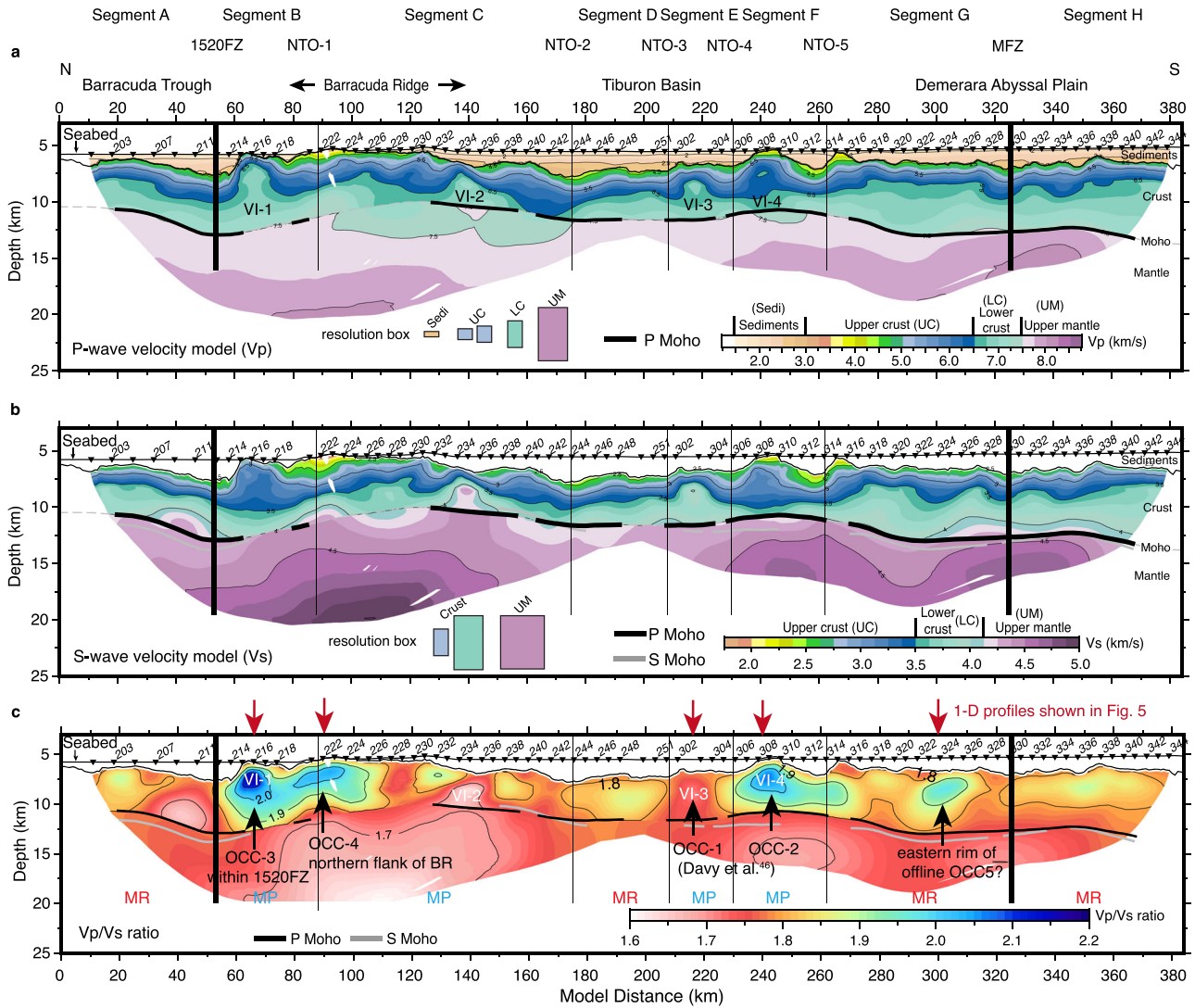

**Fig. 2 | Final travel time tomographic results. a** Final P-wave velocity (Vp) model. **b** Final S-wave velocity (Vs) model. **c** Vp/Vs ratio. All models are masked by S-wave ray coverage. Checkerboard tests reveal Vp has a resolution of 5 × 0.5 km in the sediments[56], 5 × 1 to 5 × 1.5 km in the upper crust, 5 × 2.5 km in the lower crust and 10 × 5 km in the upper mantle. Vs resolution is coarser due to larger picking uncertainty and lower ray coverage, being 5 × 2.5 to 10 × 5 km in the crust and 15 × 5 km in the upper mantle (Supplementary Table 1 and Supplementary Figs. 14, 15). Monte-Carlo analysis shows Vp uncertainty is generally <0.05 km/s in the upper crust, increasing to 0.15 km/s near the crust-mantle boundary. Vs uncertainty remains <0.05 km/s at the crustal level (Supplementary Fig. 16). The P Moho is the Moho reflector resolved from PmP phases, while the S Moho is from SmS phases. VI velocity inversion, OCC oceanic core complex, MR magma-robust segment, MP magma-poor segment, 1520FZ fifteen-tweety fracture zone, MFZ marathon fracture zone. The numbered black triangles on the seabed are the 86 ocean bottom seismometers (OBS) receiving seismic signals travelling back from the subsurface (locations shown in Fig. 4a). The classification of magma-robust and magma-poor segments is determined through a K-means cluster analysis using velocity attributes from the Vp and Vs models (see 'Methods' and Fig. 3). Note VI-3 and VI-4 were recognised as OCCs by Vp only in ref. 46 but VI-3 does not display Vp/Vs >1.9 and is therefore not serpentinised, highlighting the need to use Vs for lithological interpretation. The same abbreviations are used in other figures unless otherwise specified.

(shown by porosity) is small and so has little effect on bulk density. We converted our Vp model to density using various standard empirical velocity-density relationships for dry rocks (see 'Methods' section; Fig. 4b). We then gave areas with Vp/Vs >1.9 a reduced density. In Fig. 4d, the first density model, without any anomalous zones, gives an overall misfit of 8 mGal, with large local misfits up to +20 mGal (indicated by yellow arrows). After adding negative density anomalies, the areas with the largest misfit are significantly reduced, with best results achieved with a drop of −0.15 g/cm³ relative to the background density, leading to an average density of 2.52 g/cm³ for these bodies.

The body with the highest Vp/Vs ratio (>2.0) is VI-1, which is ~10 km wide and 2 to 3 km thick. It is located less than 1 km below the top basement and the basement above forms a dome-like structure

which outcrops at the seabed (Fig. 4a). In the gravity, the reduced density model fits the free-air anomaly better than the standard model in this region (Fig. 4c, d). The Vp/Vs geometry suggests two circular features, and we label these OCC-3 and OCC-4, which may be associated with 1520FZ and NTO-1, respectively.

Moving south, VI-2 and VI-3 do not correspond with either a Vp/Vs or a gravity anomaly. These regions might therefore contain mafics from the lower crust or ultramafics from the upper mantle exhumed to the crustal level, but not significantly hydrated or with more fractures. Alternatively, perhaps the 2D seismic line only cuts through the margin of their 3D structure. Note that VI-3 was previously interpreted as OCC-1 from Vp-information alone by ref. 46. This example highlights the need for Vs and density for a more definite interpretation.

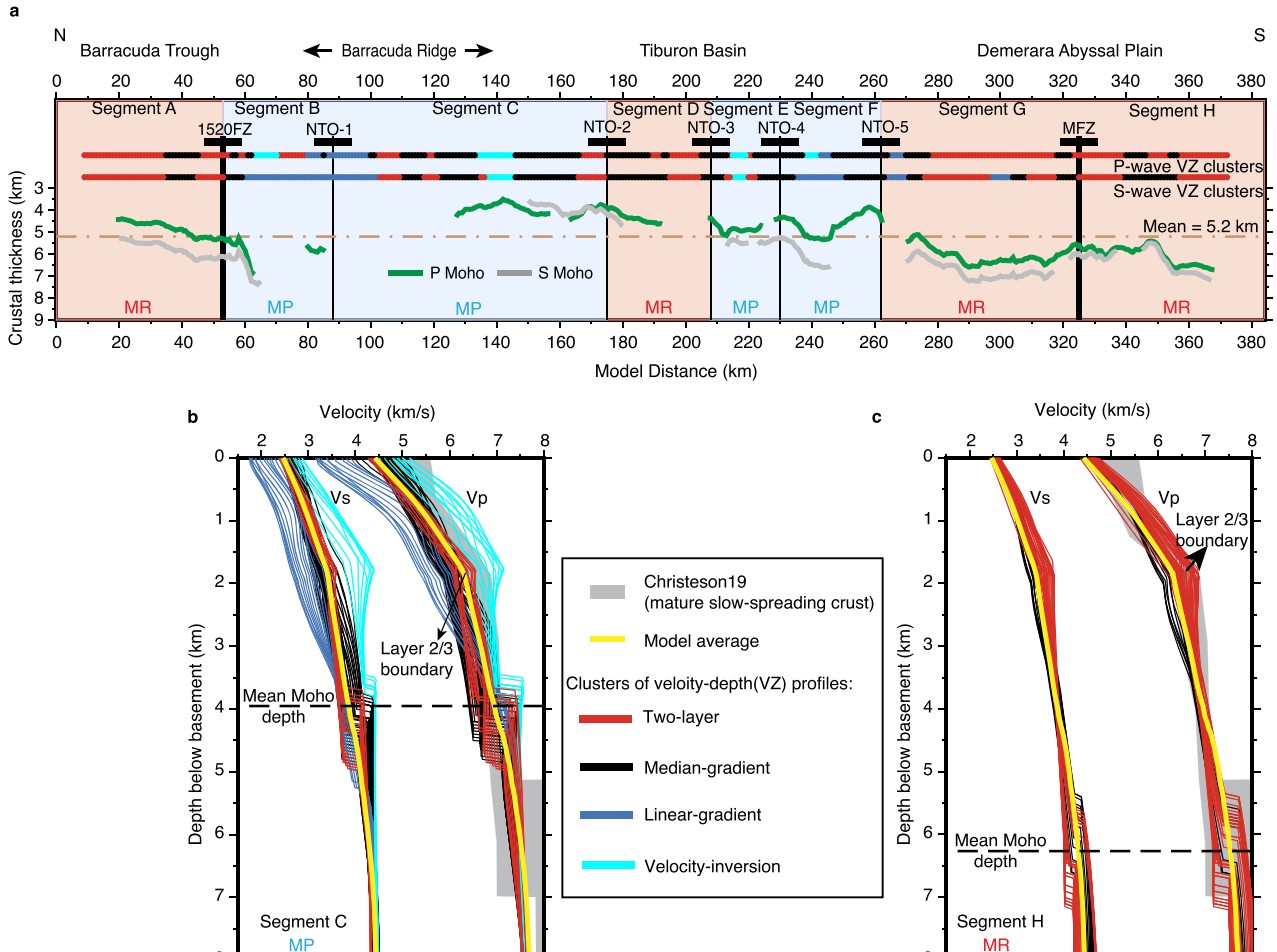

**Fig. 3 | The classification, crustal thickness and 1D velocity-depth profiles of magma-robust (MR) and magma-poor (MP) segments. a** Crustal thickness calculated by subtracting the depth of the crustal-mantle boundary, i.e. Moho interface, determined by PmP (green) and SmS (grey) reflections from the top of the basement. The classification of magma-robust and magma-poor segments is based on the distribution of four P- (Vp) and S-wave (Vs) velocity-depth (VZ) clusters (red, black, blue, and cyan bars). **b**, **c** are examples of 1D velocity-depth clusters (solid thin coloured lines) for the magma-poor segment C and the magma-robust segment H. See 'Results' section for the terminology for these clusters of velocity-depth profiles. The thick yellow line represents the average through the entire P- and S-wave velocity models. A dashed horizontal line represents the average depth of the resolved PmP reflector in each segment. The grey shade in the background is the velocity envelope from ref. 11 for mature (>7 Ma) magmatic slow-spreading crust.

Finally, VI-4 is similar to VI-1 in terms of Vp/Vs and gravity characteristics, and we label it OCC-2. The Vp/Vs model highlights a fifth region at model distances 290 to 320 km, where Vp/Vs >1.9 is present at mid-crustal depth but does not show similar P-wave velocity-depth inversions as the others. Unfortunately, it is too small to allow its characterisation with gravity. The body could be the eastern toe of a ~20 × 20 km large basement outcrop sitting west side of the seismic line (Fig. 4a). We refer to it as another potential oceanic core complex, OCC-5.

**Comparison with known oceanic core complexes**
The P- and S-wave velocity-depth profiles of the identified OCCs fall into velocity envelopes of known OCCs at slow- and ultra-slow-spreading ridges (Fig. 5), validating our interpretations. They have structural variations which might be linked to lithology that can be clearly distinguished from their seismic properties. Overall, S-wave constraints from known OCCs are much more limited than those for P-waves. However, the P- and S-wave velocity-depth profiles of OCC-4 align with the lower range of known velocity envelopes, while OCC-2 and OCC-3 align with the upper range. The eastern rim of offline OCC-5 has a velocity-depth profile that sits between those of OCC-4 and

OCC-2/OCC-3, exhibiting a two-layer structure characteristic of a magmatically robust crust.

The most prominent variation within the OCCs is the higher initial Vp gradient with a lower Vs gradient found in OCC-2 and 3 compared to OCC-4, resulting in the highest Vp/Vs ratio found on the line (Fig. 2c). Their high Vp within the top 1 km aligns with velocities of gabbroic cores from Atlantis OCC (Fig. 5). We interpret the inflexion at 1 km depth as a detachment fault zone[49], below which Vp and Vs remain high, consistent with a mixed composition of gabbroic and/or exhumed mantle observations[30,49,52,53]. As mantle exhumation by oceanic detachment faults is not fully amagmatic[18–20,54], above the inflexion, high-velocity materials likely represent shallow magmatic intrusives in the hanging wall, while the footwall may contain exhumed mantle with gabbroic plutons[24,55]. OCC-4, in contrast, has the lowest Vp and Vs and linear gradient in the top 2 to 3 km with no velocity inversions, following that of the drilled serpentinised peridotite from OCCs on MAR at 13°N and Atlantis Massif (Fig. 5). If ascribing velocity variation to lithology, OCC-4 may be predominantly composed of highly serpentinised peridotite with little or no involvement of mafic rocks.

The eastern rim of the potential offline OCC-5 stands out as a buried feature with a high Vp/Vs ratio in the centre of a

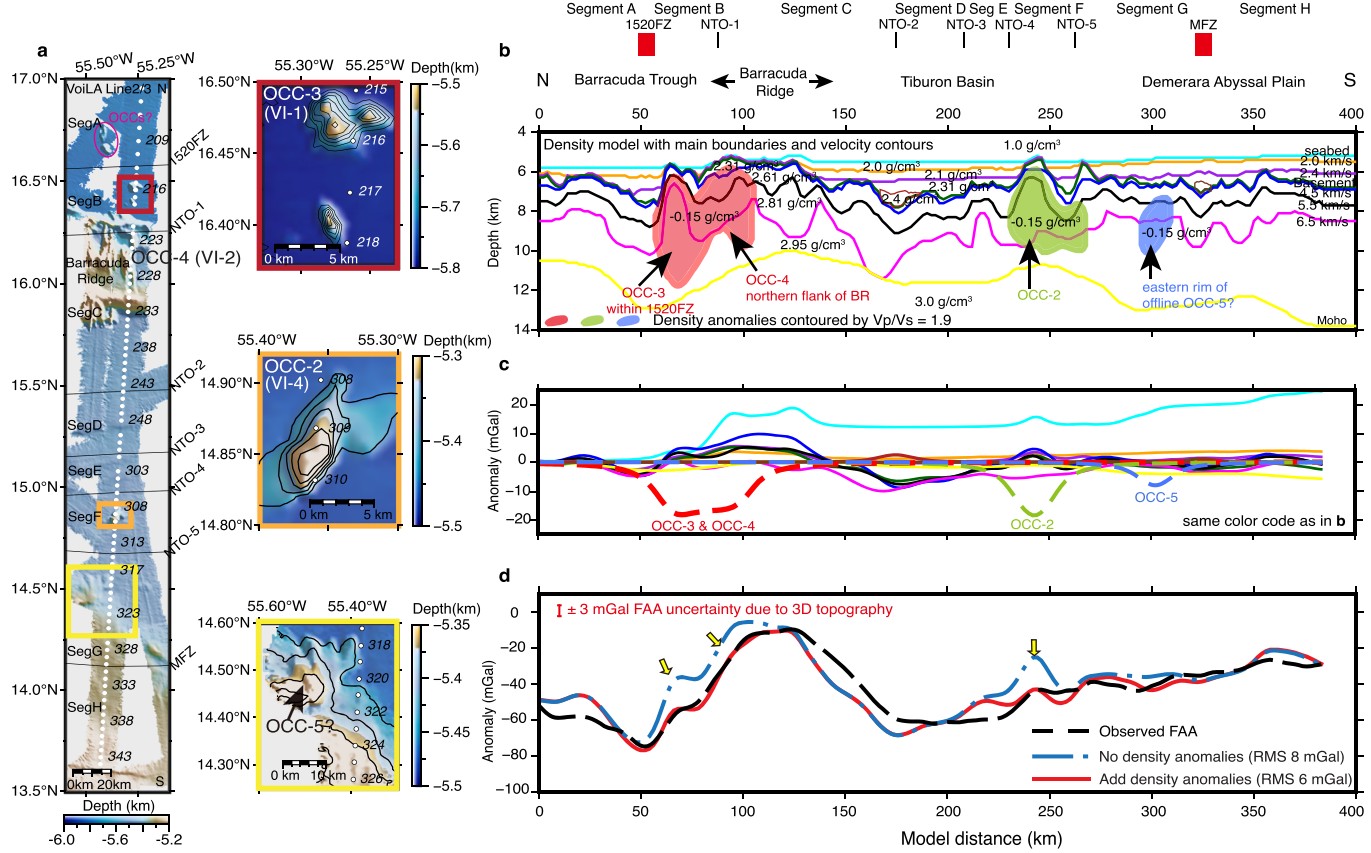

**Fig. 4 | The examination of the presence of serpentinised mantle using bathymetric maps and gravity modelling. a** Kongsberg–Simrad EM120 multi-beam bathymetry showing ocean bottom seismometer (OBS) locations (white circles) and blow-ups for basement exposures interpreted as oceanic core complexes (OCCs) with bathymetric contours every 50 m. Note the identified OCC outcrops are about 200 to 300 m higher than the surrounding seafloor, which is blanketed by 1.5 km thick sediments (Supplementary Fig. 7). **b** Density model derived from the P-wave velocity model. Cyan−seabed; orange−2.0 km/s contours; purple−2.4 km/s contours; dark red−2.8 km/s at model distance 175 km; brown−2.6 km/s at model distance 325 km; dark green−basement top; blue−4.5 km/s; black−5.5 km/s; magenta−6.5 km/s; yellow−seismically determined Moho boundary. Light pink,

light green, and light blue shades are crustal density anomalies defined by Vp/Vs = 1.9 contours. The best fit to the observed gravity occurs when they are given a negative density contrast of −0.15 g/cm³, resulting in an average density in these anomalous bodies of 2.52 g/cm³. **c** Calculated gravity anomalies for each density block with the same colour code employed as (**a**). **d** Observed free-air gravity anomalies (FAA) (black dashed line, V29.1)[47] and calculated gravity anomalies with (red solid line) and without (blue dashed line) negative density anomalies. Note the improvement in fit for inclusion of the low-density (serpentinite) bodies indicated by yellow arrows. We assign an observational uncertainty of ±3 mGal to account for 3D bathymetry on the 2D models (Supplementary Fig. 27).

magma-robust segment. This high Vp/Vs ratio is likely due to the presence of olivine-rich components in mafics to form low-velocity serpentine and/or a large proportion of other high-water hydrous phases like chlorite[13]. The proximity of the crust to the topographic high suggests it was accreted during a transition from magma-poor (west of the seismic line) to magma-robust (east of the seismic line) modes. The mafics in the magma-robust segment may have been altered by fluids from nearby outcrops, facilitated by high porosity in the basaltic layer[56]. For simplicity, we ascribe the high Vp/Vs ratio of OCC-5 to a similar hydration mechanism as other identified OCCs and thus include it in the water calculations.

## Lithology, serpentinisation, fractures and crustal physical properties

In order to generate a systematic geological interpretation of the seismic line, especially for regions with Vp/Vs >1.9, which may contain excess bound water due to serpentinisation, i.e. in addition to the typical Penrose hydration model, we evaluate the effect of serpentinisation and fracturing on Vp/Vs. First, we establish a Vp/Vs versus Vp template for each process and then project velocity measurements from our model (Fig. 2c) onto them.

The lithology/serpentinisation template was constructed from laboratory measurements of oceanic rock samples (Fig. 6a and 'Methods'). The template separates mafic and dry ultramafic rocks (dark red, mustard yellow and olive green) from serpentinised peridotite (blue and purple). The serpentinisation degree, i.e. serpentine-bound water content, increases with the increase of Vp/Vs.

The water-filled fracture template was constructed using differential effective medium theory (Fig. 6b, Supplementary Fig. 31 and 'Methods'). To simulate fracturing within the interpreted OCCs, we add randomly oriented, water-filled fractures, with aspect ratios of 0.0001–0.05, to a peridotite and gabbro medium and assess the effects on physical properties with increasing porosity. We found that for Vp/Vs >1.9 and Vp of ~7 km/s, fractured peridotite with thinner cracks (aspect ratios of 0.0001–0.002) are required while thicker cracks (aspect ratio >0.005) can explain Vp/Vs >1.9 and Vp of ~5 km/s. An increase in fracture water directly links to porosity, reducing the bulk modulus but not the shear modulus of the rock, and so primarily affects Vp. In contrast, an increase in serpentine-bound water, altering rock's mineralogy, more strongly lowers shear modulus than bulk modulus and so primarily drops Vs but less for Vp, and thus increases Vp/Vs.

Here, it is also noted that in a 65 Ma lithosphere, the high Vp/Vs anomalies present underneath over 5 km water column and sediments,

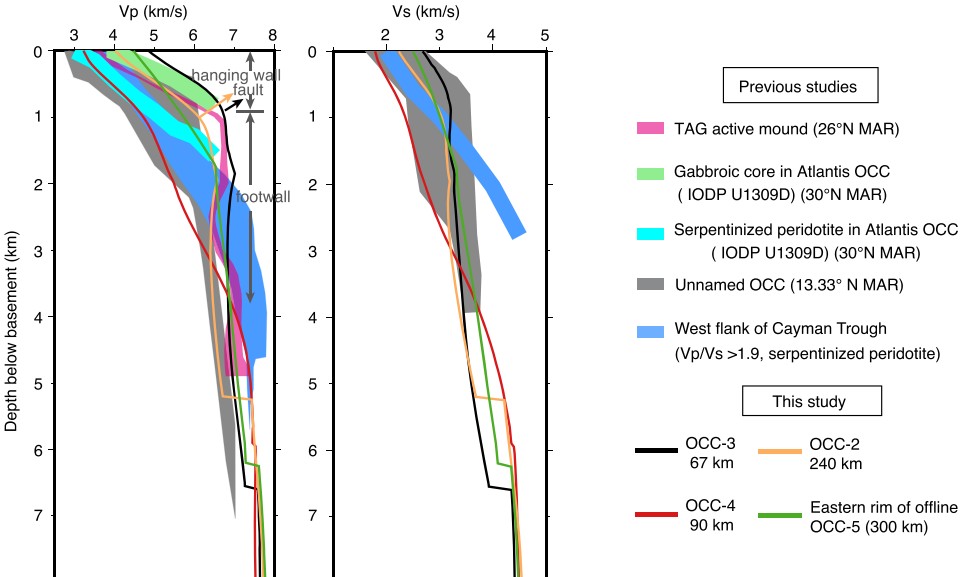

**Fig. 5 | Comparison of seismic properties through the four interpreted OCCs on the study line to those at other locations. a** P- and S-wave velocity-depth profiles (solid lines) extracted from the OCC locations along the line (for location see Fig. 2c). Shaded areas are velocity envelopes from known OCCs as follows: pink—TAG active hydrothermal mound[49]; light green/cyan—gabbros/serpentinised peridotites drilled at Atlantis[54]; grey—Unnamed OCC at 13.33°N on Mid-Atlantic Ridge (MAR)[52]; light blue—serpentinised mantle identified by Vp/Vs>1.9 at west flank of Cayman Trough[30]. The four identified OCCs have distinct features correlated with their lithological variations.

-1.5 to 2 km below top basement and extend to bottom of the crust, where fracture/cracks might be largely sealed due to sedimentary loading, increasing lithostatic pressure and precipitations of hydro-thermal mineral as lithosphere ages[11]. Moreover, maintaining free water within fractured peridotite without alteration is likely unrealistic, especially given the reactivity of seawater with olivine and the wide-spread occurrence of serpentinite in the oceans[57–59]. Therefore, we think reality is much closer to alteration than the pure fracturing model. Moreover, density variations allow us to further differentiate between the two hydration mechanisms (Fig. 5c). We rule out the pure fracture water model as the fracture porosities required to explain Vp/Vs >1.9 anomalies only yield a negligible density reduction of 0.1 g/cm³, inconsistent with our gravity results. In contrast, serpentinisation at the inferred hydration levels readily accounts for the observed density deficit of -0.5 g/cm³, supporting a serpentine-bound water-dominated system.

### Preferred lithological and excess hydration model

As previously noted, the serpentinisation model is our preferred interpretation based on the geometry and internal structure of the anomalous Vp/Vs bodies, their proximity to elevated or exposed basement and gravity signature (Figs. 4–6). Using the lithology/ser-pentinisation template (Fig. 6a), each distance/depth (x, z) point from the Vp/Vs model (Fig. 2c) was categorised to give the lithological and excess bound water model due to serpentinisation shown in Fig. 7. It should be noted that this is an upper bound estimate of the excess water content. We present two alternate models, one with a combi-nation of serpentinisation and fracturing and one with pure fracturing in the Supplementary Figs. 31, 32. Note that water stored in other forms of hydrous minerals due to alteration and/or pore-clogging, especially in the upper mafic crust, is not included in the calculation (see 'Methods' section). Instead, we assume the Penrose hydration model applies to magma-robust and other portions of magma-poor segments (white regions of the upper panels in Fig. 7b).

Figure 7 shows that in magma-robust segments, basalts overlying dolerite/gabbros in the upper -2 km are estimated to have negligible excess hydration with the exception of the eastern rim of OCC-5. In other words, according to our classification, the hydration levels of these segments do not surpass what would be expected for Penrose-style oceanic crust of this age. In magma-poor segments, the boundary between basaltic and gabbroic sections is laterally heterogeneous and interrupted by regions with higher excess bound water content (at model distances 50 to 110 km and 230 to 270 km). These hydrated pockets overall give a bound water content estimate of 4 wt% averaged across the crustal column.

Below the Moho, the mantle is effectively dry throughout the line. The seismic properties are nearly identical to those on our orthogonal line before bending[45] and so anisotropy does not seem significant. In magma-robust segments, the dry mantle consistently has Vp/Vs over 1.70, perhaps representing a more depleted mantle. However, Vp/Vs is notably below 1.70 in magma-poor segments, suggesting a more pri-mitive mantle due to limited melting. There are a few potentially interesting anomalies in the magma-poor segment C below Barracuda Ridge (model distances 90 to 170 km), with dry mafics displaced within the mantle portion. However, as they are not central to the water estimation due to serpentinisation in the crustal column, we will not discuss them further in this paper.

## Discussion

Our results provide detailed seismic properties across eight slow-spreading segments on the incoming plate of the LASZ. We further discuss the amount and distribution of hydration and its implications for Atlantic subduction zone processes.

The identified OCCs occupy about 17% of the total crustal column, and dominantly in magma-poor segments. These local 'hydrated pockets' have bound water content up to 12 wt% at depths of -2 km below top basement (Fig. 7b). Averaged over the entire line, we estimate that the heavily serpentinised regions, i.e. considered as composed of serpentine (Vp/Vs >2 and water con-tent of 10 to 13 wt%) form 3.4% of the area. This is lower but of a similar magnitude to the 6% value from ref. 24 based on obser-vations at the ridge axis. According to the estimation strategy from ref. 26, a cubic metre of serpentinite with a density of 2.52 g/cm³ would contain 252 to 328 kg of water. 1% of a 5.2 km

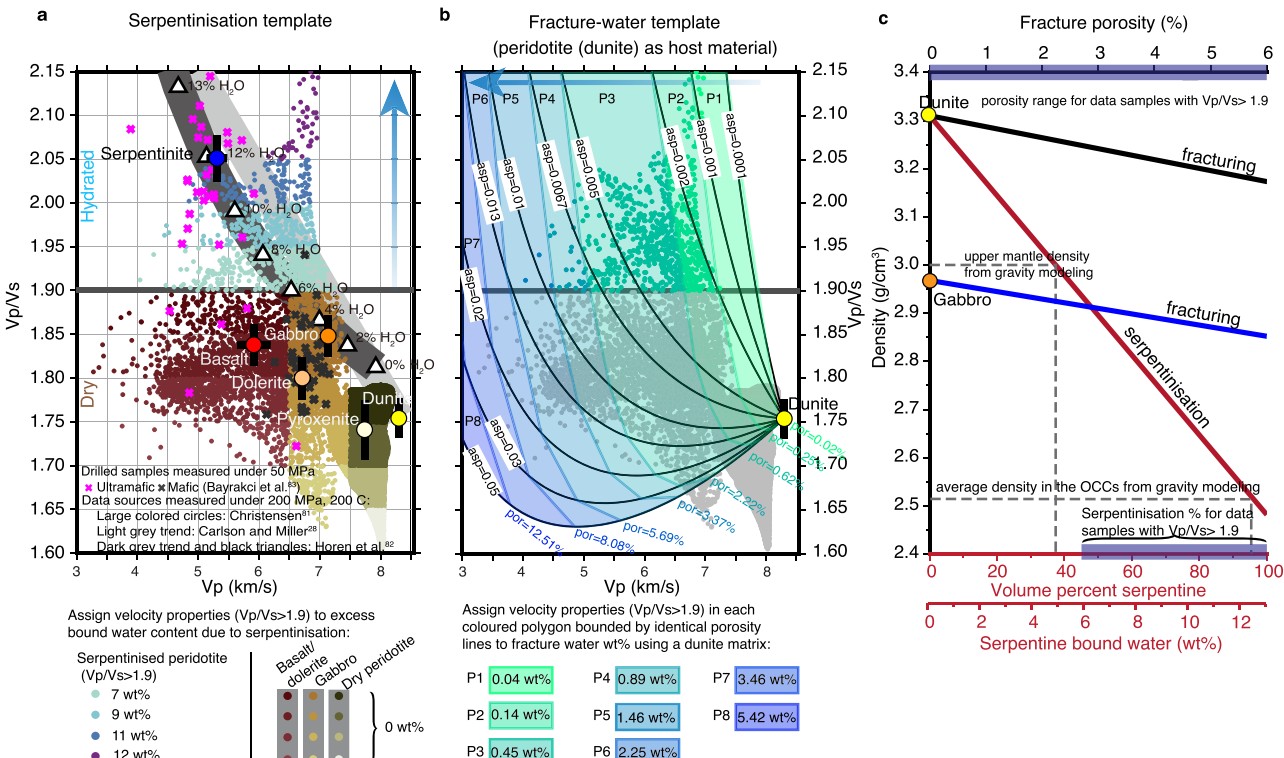

**Fig. 6 | Interpretation of Vp/Vs anomalies. a** Lithology−serpentinisation−Vp/Vs relationship. Velocity points (small filled circles) from final velocity models (Fig. 2) are plotted on a Vp/Vs versus Vp lithological template. Lithological bounds ('Methods') are based on laboratory measurements from Christensen[81] (large coloured circles), drilled samples[83] (magenta/dark grey crosses) and serpentinisation trends from peridotite to serpentinite (light grey−ref. 28; dark grey−ref. 82). White triangles indicate bound water content[82] with the blue arrow showing increasing bound water content with higher Vp/Vs. Dark red/mustard yellow/olive green velocity points represent standard crustal/mantle rocks, so assigned 0 wt% excess bound water content. Blue/purple velocity points are assigned increased bound water content ('Methods' and Supplementary Table 2). The classified velocity points appear in lithological model positions (Fig. 7a), with corresponding excess bound water content in Fig. 7b. **b** Water-filled fractures versus Vp/Vs. The

small filled grey circles are the velocity points as shown in (**a**). Black lines display Vp/Vs-Vp trends for water-filled fractures with different aspect ratios (asp) calculated from differential effective medium (DEM) analysis with dry dunite as the host material ('Methods'). The coloured polygons delineate regions with different fracture porosities (and hence water content) (Supplementary Table 3). The blue arrow shows that fracture water increases as Vp decreases. The classified velocity points (Vp/Vs >1.9) are shown in their model positions with the converted excess bound water content in Supplementary Fig. 32b. **c** Bulk density changes with dunite/gabbro porosity (top axis) and serpentinisation degree of dunite (bottom axis). Note that the density reduction associated with the fracture porosity range for Vp/Vs >1.9 is insufficient to account for the density drop modelled from gravity (Fig. 4). However, this density reduction is easily achieved by serpentinisation at levels indicated by velocity points with Vp/Vs >1.9.

average crustal column amounts to a 52 m column of serpentinite, storing $1.3 \times 10^4$ to $1.7 \times 10^4$ kg of water per m² of crust. Therefore, an average of 3.4% serpentine would then contain $4.4 \times 10^4$ to $5.8 \times 10^4$ kg of water per m² of crust. For comparison, ref. 26 estimated an average of $4 \times 10^4$ kg per m² water contained in pore spaces of 'Penrose-style' oceanic upper crust. Thus, these highly serpentinised regions, although only small in volume, must be taken into account when developing a water budget for subduction zones.

Many previous studies have demonstrated that fracture zones, especially large-offset ones, are sites of increased hydration due to water percolating into faults within the transform fault zone[15,60,61]. In our study region, however, the MFZ shows no signs of additional hydration at our resolution scale, while there are strong hydration signals near several of the second-order NTO discontinuities and within magma-poor segments. Thus, our results suggest a direct link between hydration and magma budget rather than ridge-axis tectonic deformation of any discontinuity. The emplacement of ultramafics during magma-poor episodes, either at oceanic transform faults, or other types of discontinuities and/or even within the ridge-spreading segments, exerts the key control on the abundance and distribution of the water storage. We suggest the pattern of hydration of the slow-spreading crust is discontinuous and disorganised on a temporal and

spatial scale, alternating naturally with magma-poor segments. Such temporal and spatial variability is simulated in recent numerical models and results from the spontaneous self-organisation of magma supply controlled by spreading rate, mantle temperature and fault weakening (e.g. by hydration) in a mechanism of tectono-magmatic instability during ridge spreading[62,63]. This is significant because when the variable 'hydrated pockets' of slow-spreading oceanic lithosphere is subducted, this will 'smear' the water delivery over time compared to the more organised fracture-zone associated hydration of fast-spreading oceanic lithosphere[64].

Previous slab dehydration models[6,17] consider only Penrose stratigraphy in the slab crust, which applies more to fast- and intermediate-spreading but not slow-spreading settings. Penrose crust stores about 1.1 wt% water when averaged across its crustal column[16] (Supplementary Fig. 33). In contrast, the OCCs identified in our profile have an excess bound water content due to serpentinisation of 4 wt% when averaged across the crustal column (Fig. 7b), four times more than that in a Penrose crust. If we consider the subducted slow-spreading Atlantic crust consists of 17% serpentinised peridotite and 83% Penrose stratigraphy, this results in an average of up to 1.6 wt% bound water, which is about 1.5 times more than that of pure Penrose crustal composition used in previous models and potentially adding a 50% increase to Atlantic subduction bound water budget estimates.

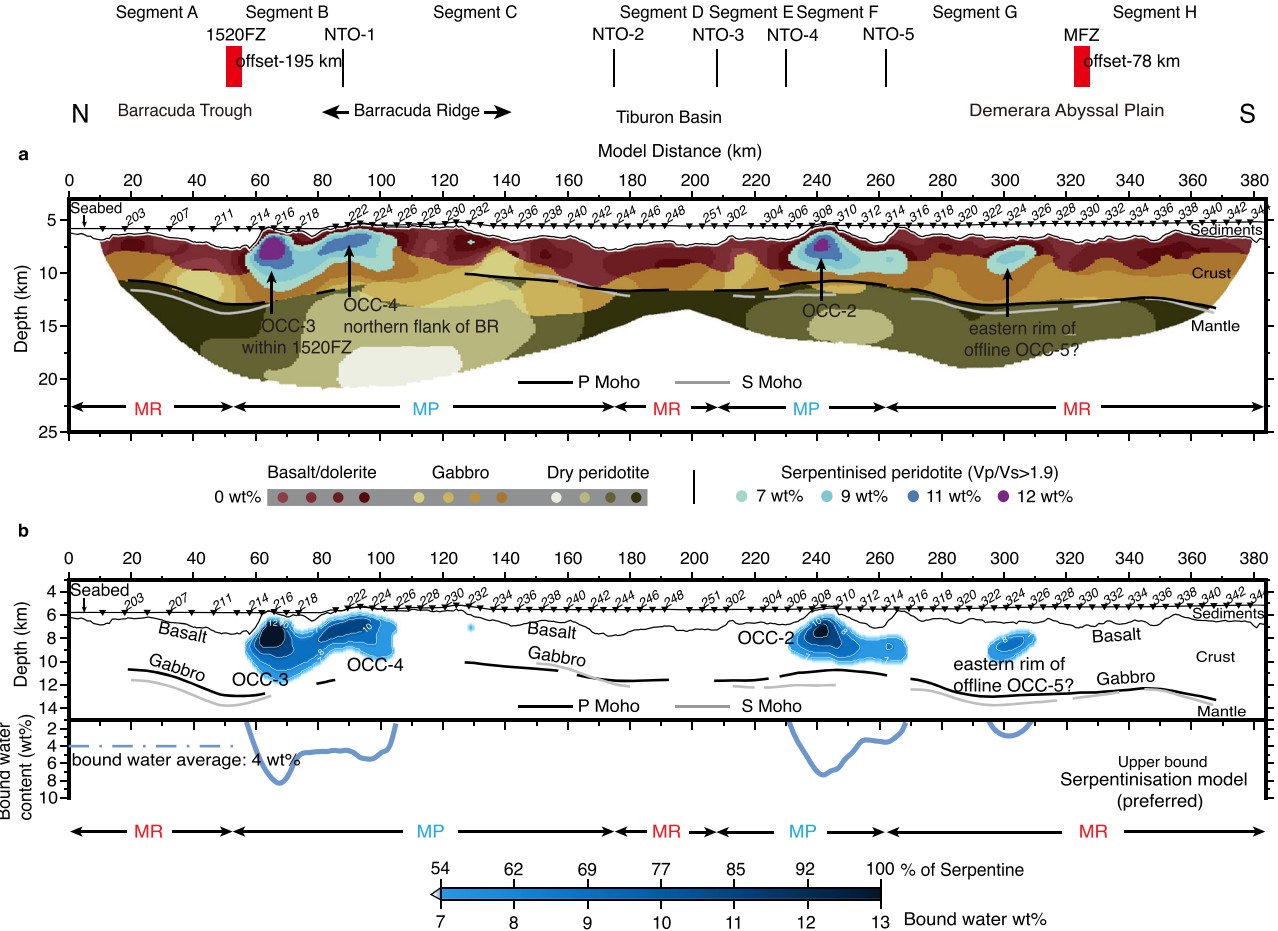

**Fig. 7 | Lithological model and estimation of excess bound water content due to serpentinisation in the mature slow-spreading crust. a** Our preferred lithological interpretation based on the template shown in Fig. 6a. Velocity points in dark red, mustard yellow, and olive green are interpreted as standard crustal and mantle rocks, while velocity points in blue and purple are interpreted as serpentinised peridotites in the form of oceanic core complexes (OCCs). Spreading segmentation and crustal accretion modes are also shown for reference. **b** Our preferred excess hydration model, where Vp/Vs >1.9 is interpreted as serpentinite. The upper panel shows the excess bound water content in the crustal column converted from Vp/Vs ratios using relations from ref. 82 (see 'Methods' section, Fig. 6a, Supplementary Table 2 and Supplementary Fig. 29). The lower panel shows the averaged excess bound water content contained in these OCCs (blue lines) over the whole crustal column at each model distance, resulting a line average (blue dashed line) at these OCCs of 4 wt%. Spreading segmentation and crustal accretion modes are also shown for reference. Note that this interpretation places an upper bound on the excess water content estimate due to serpentinisation.

Our results indicate that before plate bending in the outer rise of LASZ, the upper 10 km of the mantle, even in the magma-poor segments, is effectively dry. Allen et al.[45] show bending-faults add 16% serpentinisation (about 2 wt% water) to the upper 4 km mantle. Combined with our results, a cumulative serpentinisation of about 28% (about 3.6 wt% water) is present in the upper 9 km Atlantic lithosphere (crust and mantle each occupy half) as it enters the subduction zone. We should note, however, that these estimates are an upper limit, as velocity reduction might also result from increased crack-like porosities by organised bend faulting[65,66].

As well as an increase in the estimate of overall hydration levels of slow-spreading oceanic lithosphere, our work shows the hydration to be highly irregular or 'patchy'. Within the LASZ itself, this observation may explain the occurrence of intraplate seismicity in clusters rather than in a continuous, linear pattern along the projection of fracture zones[67,68]. The hydrated patches occur within magma-poor segments, and our ridge perpendicular line showed modes of seafloor spreading lasting at least 3–5 Myr[45]. Hence, our work predicts that the delivery of water into the LASZ will vary greatly in space and time. Given the dehydration of subducted serpentines has been proposed to significantly contribute to earthquake rates and volcanic productivity in the LASZ[67–69], the delivery of these hydrated pockets is likely to impart

major control on subduction zone behaviour and history. Note that our results can be further included in the calculation of the overall water budget for the subducted Atlantic lithosphere, leading to a better understanding of slow-spreading crust subduction and its role in the global water cycle. Our results also have implications for global estimates of water budgets back through geological time during supercontinent breakup, when slow-spreading crust subduction was more common than today[25].

## Methods

### Seismic data acquisition and preprocessing

The wide-angle seismic data were acquired by 86 4-component ocean bottom seismometers (OBS) spaced ~4.5 km apart, with 45 OBS deployed in two groups (hence line 2/3). Instruments were shot at 60 s (~150 to 190 m), and the northern set was additionally shot at 20 s (50 m) during a multi-channel seismic line acquisition. Eleven instruments in the centre of the line recorded both the northern and southern shots. Seismic data acquisition parameters are detailed in ref. 40. Each OBS record was band-pass filtered (zero-phase) with corner frequencies of 3, 5, 20 and 30 Hz. For the P-wave inversion, a similar strategy to ref. 46 was adopted, with the advance that shallow sedimentary and upper crustal velocities were improved using

downward continued (DC) OBS records[56]. For the S-wave inversion, first, a hodogram analysis of the direct wave arrivals was employed to determine the OBS orientation of horizontal (X and Y) geophone components in relation to the shot line (Supplementary Fig. 1). These components were then rotated to give radial and transverse directions[36], with the radial direction providing the highest signal-to-noise ratio for S-wave picking.

## Identifying converted S- and P-wave phases

Multiple phases on OBS records were identified by preliminary forward modelling with the RayInvr software[70] and TOMO2D codes[71] (Supplementary Figs. 2, 3). For S-wave phase identification, an initial Vp/Vs ratio of 1.8 in the crust and upper mantle was assumed. Mid-crustal P-wave refractions outside the water wave cone are nearly subhorizontal, with an apparent velocity of ~8 km/s (Supplementary Fig. 3a, c), while converted S-wave refractions have an apparent velocity of ~4.5 km/s (Supplementary Fig. 3b, d). There is a family of S-wave arrivals, as conversion can happen at the sediment-basement interface on the way down, on the way up or both. The first group, termed Sg, Sn, and SmS (conversion on way down), are only observed on the geophone components (vertical, radial and transverse) but not on the hydrophone component (Supplementary Fig. 2). Sg was picked for offsets below 40 km, Sn from 40 to 160 km and SmS from 20 to 50 km (Supplementary Fig. 3c, d). The second group, PPS phases (PPgS and PPnS) travel as P-wave in the crust and uppermost mantle but convert to S-wave at the sediment-basement interface underneath OBS on the upward propagating leg. They have travel time-distance branches that are parallel to Pg/Pn with delayed arrival times in geophone components (Supplementary Fig. 2b). They were picked to estimate the delay time of the S-wave relative to the P-wave travelling in the sediments below each instrument. This delay was subtracted from Sg/Sn/SmS arrivals to produce 'effective PSP arrivals'[36] i.e. as if they had travelled through the sediments at both source and receiver sides as P-waves but through the crust/mantle as S-waves in order to give symmetry for tomographic inversion using Vp in the sediments (Supplementary Fig. 4). The final group of S-waves, PSP phases, including PSgP, PSnP and PSmSP, convert on both the way down and up, so they only travel below basement as S-waves. These double-mode converted phases are only intermittently observed on the hydrophone and vertical components (Supplementary Fig. 2b) and were not used in this analysis.

S-wave data quality was categorised as good (45% of OBS records), moderate (38%) or poor (17%) based on signal-to-noise ratio and continuity of arrivals for travel time picking (Supplementary Fig. 5). Picking uncertainties for the S-wave arrivals were assigned based on their offset, with a starting uncertainty of 0.75, 1.0, and 1.25 T for good, moderate and poor arrival quality respectively (Supplementary Table 1). T is 200 ms, which is the average period of the S-waves for all offset ranges.

For P-waves, crustal refractions (Pg2 and Pg3) in the upper and lower crust (Supplementary Fig. 3a, c) are seen up to 40 km. They change from low to high apparent velocity with variable amplitude. Pg2 turning within the top 2 km below the basement inside the water wave is shown and confidently pickable on DC OBS records[56]. Mantle refractions (Pn) show low amplitude with an apparent velocity above 7.5 km/s at offsets up to ~150 km. Moho reflections (PmP) are spatially limited to offsets of 20 to 40 km and observed on ~80% of OBS records. Picking uncertainties were assigned based on offset for Pg and Pn (Supplementary Table 1)[46]. For PmP phases, 50, 75, and 100 ms were respectively assigned based on arrival continuity with high-, mid- and low-level of picking confidence (Supplementary Fig. 6 and Supplementary Table 1). In comparison, SmS phases are present at similar offset ranges but with weaker amplitudes.

## Travel time tomographic inversion of P- and converted S-waves

Vp was inverted using TOMO2D[71] in a layer-stripping procedure[72] by adding data sequentially: (1) first for the sedimentary model using Ps and basement reflections (PbP) picked from DC OBS records[56], (2) next for the upper crustal model using Pg2 phases picked from DC OBS records while fixing the sedimentary model[56], (3) next by combined inversion of full crustal model and the Moho depth using Pg and PmP phases from original (non-DC) 60 s records, and (4) finally for upper mantle model using Pn phases while leaving the crustal model damped to maintain minimum changes.

The final P-wave sedimentary model (Supplementary Fig. 7) was fixed in the subsequent steps, below which a starting model guided by 1D velocity-depth relations for slow-spreading mature crust from ref. [11] was defined with a top basement velocity of 4.5 km/s[56] (Supplementary Fig. 8a). The grid spacing for the inversion was $\triangle x = 500$ m and $\triangle z = 50$ m constantly in the upper crust (~2 km below basement) and then increased linearly to $\triangle z = 500$ m at the base of the model. The smoothing correlation lengths increase from 3 to 5.2 km horizontally and 0.5 to 1.5 km vertically from 5 to 25 km depth. A depth weight kernel of 1 was used, giving an equal weighting between velocity and Moho depth perturbations. In each step, variable damping was applied, defined by ray coverage of the previous model step, to allow velocity perturbations favouring deeper model regions[46].

The starting Vs model was derived from the final Vp model using a Vp/Vs ratio of 1.8 below the basement (Supplementary Fig. 8b). The same inversion strategy was applied for both P- and S-wave tomography. First, we applied downward continuation to the radial components (Supplementary Fig. 9). However, this process is less essential than for Vp modelling, as weak water waves on radial and transverse components do not affect S-wave picking at short offsets. SmS phases were picked from less than 30% of OBS records (Supplementary Fig. 6), so the Moho depth was constrained by PmP travel times and fixed in the Vs inversion. The Moho depth was then inverted as a final step using SmS for comparison with the P Moho. Final Vp and Vs models show $\chi^2$ close to 1 with reduced RMS misfits between the picked and modelled travel times (Supplementary Fig. 10).

## S-wave velocity model inversion using different starting models

To assess the independence of our final S-wave velocity model from the selection of the starting model, we conducted a comparison test using a starting model without prior information about the final P-wave velocity. This starting model is purely from the conversion of the initial P-wave 1D velocity-depth profile into S-wave using a Vp/Vs of 1.8 (Supplementary Fig. 11) and the same initial Moho reflector as Vp modelling. The same inversion strategy and modelling parameters were applied.

The final inverted S-wave model from the simplified starting model (Supplementary Fig. 12b) shows high similarities in ridge-parallel (along-strike) Vs velocity variations with the preferred model (Supplementary Fig. 12a), except for localised differences at topographic highs (at model distances 80 to 120, 240, and 265 km). These differences are possibly due to relatively lower ray coverage of S-waves beneath these basement highs (Supplementary Fig. 14) caused by poor P-to-S conversion at these locations. Without prior Vp-information, the low-velocity anomalies beneath these basement highs are unable to be recovered. The Vp/Vs model (Supplementary Fig. 13) also shows similar patterns despite some localised differences. This indicates that the choice of starting models does not significantly impact the modelling outcome, confirming the reliability of our interpretation based on the preferred Vs model.

## Quality analysis and checks for final P- and S-wave models

Following ref. [46], we conducted quality analysis for the final inverted velocity models. In general, rays of P-waves are about double that of S-waves. The derivative weight sum (DWS) shows overall good ray

coverage (>10 rays per cell) in the crust and upper mantle except for a lack of ray coverage underneath Barracuda Ridge for the S-wave model (Supplementary Fig. 14). Checkerboard tests performed with a range of velocity anomaly sizes at an amplitude of ±5% and with a gaussian noise level (standard deviation of 10, 50, and 100 ms) scaled by the pick uncertainty for P- and S-waves, reveal a resolution for Vp of $5 \times 1$ km to $5 \times 1.5$ in the upper crust, $5 \times 2.5$ km in the lower crust and $10 \times 5$ km in the upper mantle, and for Vs of $5 \times 2.5$ to $10 \times 5$ km in the crust and $15 \times 5$ km in the upper mantle (Supplementary Fig. 15).

In the Monte-Carlo analysis, we randomised the velocities at the top and bottom of the upper, lower crust and top mantle by ±5% of the starting model (Supplementary Fig. 8) with the thickness of the total crust randomised by ±1 km for P-wave inversions. The final Moho reflector inverted from PmP was used and remained unchanged in the S-wave starting models.

Velocity uncertainty of P-wave tomography (Supplementary Fig. 16a) is <0.02 km/s in the sediments, and up to ±0.05 km/s but mostly <0.03 km/s in the upper crust. This uncertainty results in an 80 to 90% reduction in the mean deviation of the upper crust, i.e. from ~±0.2 to ±0.3 km/s (assuming upper crustal velocity of 4.5 to 6.5 km/s with ±5% deviation) in the starting models to -0.03 km/s deviation in the final models. In the lower crustal depth (between 2 to 5 km below the top basement), Vp uncertainty is ~± 0.07 km/s with peaks at ±0.17 km/s (mean deviation reduced by 30%). This suggests the final model's resolution is minimally impacted by starting models, resolving velocity within the estimated uncertainty bounds[73]. The depth uncertainty for the P Moho reflector resolved from PmP phases is ±0.5 km, corresponding to a region of the highest velocity uncertainty up to ±0.25 km/s. In the mantle, Vp uncertainty ranges from ±0.07 to ±0.13 km/s in the south, reducing to -0.07 or <0.05 km/s at ~17 km depth below the basement under Tiburon Basin, Barracuda Ridge and Barracuda Trough in the north.

The S-wave model shows uncertainties generally below 0.05 km/s (Supplementary Fig. 16b). As a fixed Moho reflector was used for each inversion, in contrast, the uncertainty across the crust-mantle boundary in the S-wave model is comparably lower and reaches a maximum of ±0.1 km/s. The Vp/Vs ratio uncertainties were calculated from the standard deviation by combining 100 different Vp and Vs models randomly (Supplementary Fig. 16c). The uncertainty is mostly <0.05 km/s in the crust with a slightly higher uncertainty (up to 0.07 km/s) below Barracuda Ridge and up to 0.1 km/s within the crust-mantle boundary zone. However, it must be noted that, even in the worst resolved areas, such as the crust-mantle boundary, the control on velocity and velocity gradients is good enough and does not affect the interpretation.

### K-means cluster analysis for grouping velocity-depth characters

Davy et al.[46] identified two distinct types of P-wave velocity-depth (VZ) profiles south of 15.5°N, which they attributed to magma-robust and magma-poor seafloor spreading modes. Their analysis was done by manual visualisation. This resulted in the categorisation of segments (Segments D to H) into one of the two modes, with the recognition that VZ profiles at segment ends were slightly modified from their centres. Velocity inversions were also noted but not formally categorised. We instead performed a K-Means cluster analysis[74,75] of the P- and S-wave VZ profiles, a more robust statistical method, to identify regions with similar velocity structures. To our knowledge, this type of analysis has not been previously applied to classify velocity-depth profiles in the oceanic crust.

To determine the optimal number of clusters (K), we used the silhouette value, which measures how similar a VZ profile is to its own cluster's centroid compared to others[76,77]. The optimal K maximises the averaged silhouette value from 20 iterations of K-means algorithms by randomising the initialisation of the cluster centroids each time. To better partition the VZ profiles, we constructed four

categories of features for Vp and Vs input to K-means: (1) VZ profiles, (2) velocity gradient, (3) minimum value of the second derivative of depth to velocity (calculated every 50 m in depth), and (4) maximum value of the standard deviation of velocity gradient calculated every 500 m in depth (Supplementary Figs. 17, 18). The last two categories are specially designed for identifying rapid velocity changes with depth. The four categories were then concatenated as one observation at locations every 1 km along the seismic line. Tests showed that the velocity gradient is the key for grouping velocity structures.

We found four clusters (K = 4) that best describe Vp and Vs structures with exclusive characters. With two clusters (K = 2), velocity inversions are unidentifiable and mixed with two-layer velocity-gradient seismic structures (Supplementary Fig. 19). Both K = 3 and K = 4 can detect velocity inversions (cyan) but K = 4 is preferred as it also distinguishes regions with the lowest velocity gradients (dark blue) (Supplementary Figs. 20, 21), possibly related to pervasive hydration. Both K = 5 and K = 6 include redundant clusters, which are not central to the interpretation (Supplementary Figs. 22, 23). All the interpretations are based on the preferred four-cluster solution (Supplementary Fig. 21).

### Gravity modelling

Our Vp model was checked for consistency against the coincident satellite-derived free-air gravity anomaly (FAA, V29.1)[47]. The uncertainties of 2D gravity modelling due to the effect of 3D topography were first assessed (Supplementary Fig. 27). The likely error in the observed gravity is ~±3 mGal. Therefore, the fit to the gravity is deemed acceptable if it matches this range.

To estimate the gravity, a 2D model with homogeneous density blocks was derived from Vp. The block shapes were based on physical boundaries (seabed, basement and Moho) and velocity contours representing lateral and vertical variations. Velocity contours of 2, 2.4 and 2.6 km/s at MFZ and 2.8 km/s at Tiburon Basin were extracted for sediments, and 4.5 to 6.5 km/s for the oceanic crust (Supplementary Fig. 28a). Density was converted from Vp using empirical velocity-density relationships for sediments[78] and oceanic crust[79]. For the mantle, we tested densities from 2.8 to 3.3 g/cm³. The predicted FAA was subsequently computed using a 2D line integral algorithm of ref. 80. The best fitting mantle density is 3 g/cm³, which was used for further analysis.

To evaluate the effect of the high Vp/Vs anomalies on the gravity modelling, we introduced three density bodies defined by Vp/Vs contours of 1.9. We then tested by forward modelling values between −0.1 and −0.2 g/cm³ to account for the drop in density that might be expected if these bodies contained significant quantities of serpentinite as opposed to water-filled fractured rock[50,73]. Given the issue of real 3D geological structures captured on a 2D profile, the aim was not to determine the best-fit density of these bodies but rather to demonstrate whether the gravity was consistent with the presence of serpentinite or not. We found that a negative density of −0.15 g/cm³ relative to the background density in the crust gave the best overall fit (Fig. 4b–d).

### Relationships between lithology, serpentinisation and Vp/Vs

Several studies[28,30,81,82] have previously proposed relationships between Vp/Vs and lithology. We re-examined these relationships in the context of our seismic model results and also a recent compilation of laboratory measurements of Vp and Vs of samples from deep ocean drilling[83]. The laboratory measurements used here to establish the lithology, serpentinisation and Vp/Vs relationship were conducted under pressure at 50 to 200 MPa[28,81–83], simulating depths of 2 to 8 km−conditions at the top basement and near the Moho. We broadly followed the classification scheme shown in ref. 30 including adopting a Vp/Vs ratio of 1.9 as the main division between mafic and

dry peridotite (below 1.9) and altered, serpentinised peridotite (above 1.9) at crustal pressures but with the following modifications (Fig. 6a and Supplementary Fig. 29). To be more consistent with our seismic results we placed the boundary between basalt/dolerite and gabbro at Vp = 6.5 km/s (as opposed to Vp = 6.6 km/s). We placed the boundary between gabbro and dry mantle at Vp = 7.5 km/s (as opposed to 7.15 km/s) to align with our seismically defined Moho interface. To estimate the excess bound water content due to serpentinisation, we focus on velocity points with Vp/Vs >1.9 only.

### Relationship between fractures and Vp/Vs

To investigate the effect of fractures on Vp and Vp/Vs, we include a differential effective medium (DEM) analysis following the methods outlined in previous studies[84,85]. We used the intrinsic properties of dry dunite (Vp = 8.299 km/s, Vs = 4.731 km/s, density = 3.310 g/cm³) and dry gabbro (Vp = 7.138 km/s, Vs = 3.862 km/s, density = 2.968 g/cm³, measured under 200 MPa, 200 °C from ref. 81) for the host material in the DEM calculations. Note that these host materials were selected based on the lithology of drilled cores of known OCCs[53,54]. We added increasing proportions of randomly oriented water-filled fractures with aspect ratio of 0.0001 (thin) to 0.05 (thick) to calculate the effective Vp, Vs and Vp/Vs with increasing of porosity (Supplementary Fig. 30). We found that the range of aspect ratios can cover all our observed velocity points (grey small circles in the background in Supplementary Fig. 31b) when a dunite matrix is used, but even with much thinner cracks (aspect ratio = 0.00001), the fracture model with a gabbro matrix cannot fully cover the velocity points with Vp/Vs >1.9 and Vp of ~7 km/s (Supplementary Fig. 31c).

### Water content calculations for anomalous crust

We set up and evaluated three kinds of possible hydration models to estimate the water content of the anomalous crust defined by Vp/Vs >1.9. In Model 1, termed as the serpentinisation model, we interpreted velocity points with Vp/Vs >1.9 as serpentinised peridotites. We used the relations linking Vp and Vs to serpentine percentage from ref. 82 and further to bound water content (large white triangles in Fig. 6a and Supplementary Fig. 29) by assuming 13 wt% water content for fully serpentinised peridotites and 0 wt% for non-serpentinised, unaltered peridotites following ref. 50 (Supplementary Table 2). This converts Vp/Vs to the excess bound water content model in Fig. 7b (also Supplementary Fig. 32a). Note that mafic rocks with Vp/Vs <1.9 are considered to be standard composition used in a Penrose hydration model (Supplementary Fig. 33) with 0 wt% excess bound water assigned and not included in our excess bound water model.

In Model 2, termed as the fracture water model, we interpreted velocity points with Vp/Vs >1.9 as either water-filled fractured peridotite (Model 2a; Supplementary Fig. 32b) or water-filled fractured gabbro (Model 2b; Supplementary Fig. 32c). The identical porosity on each Vp/Vs versus Vp trend defined by different aspect ratios is connected and used to divide regions (polygons) of different fracture-water weight content (Fig. 6b, Supplementary Fig. 31b, c and Supplementary Table 3). Velocity points falling into the polygons defined by neighbouring porosity lines are assigned the averaged fracture-water weight content (Supplementary Table 3), converting Vp/Vs to the fracture-water models. The result is shown in Supplementary Fig. 32b, c.

In Model 3, termed the hybrid model, both serpentinisation and fracturing are considered. Velocity points with Vp/Vs >1.9 that lie on the serpentine hydration trend are interpreted as serpentinised peridotite, and those that lie off this trend as fractured peridotite. Velocity points are assigned either excess bound water or fracture water based on rules described in Model 1 and Model 2a (Supplementary Figs. 31d, 32d). Note that we choose only fractured peridotite as an example to set up the hybrid model as the fracture-water template of a peridotite matrix can cover all velocity points with Vp/Vs > 1.9 and also the overall fracture water content estimated based on different rock matrices is similar (Supplementary Fig. 32b, c) which therefore has little effect on the hydration estimation in a hybrid mode.

### Data availability

Bathymetric data, seismic navigation and relocated OBS records of four components are available from the Marine Geoscience Data System (https://www.marine-geo.org/tools/search/entry.php?id=JC149) under the condition of acknowledging ref. 46 and this paper. The final inverted Vp (Supplementary Data 1), Vs (Supplementary Data 2), and Vp/Vs ratio (Supplementary Data 3) grids are uploaded as Supplementary Information.

### Code availability

Code for travel time tomography TOMO2D is available from http://people.earth.yale.edu/software/jun-korenaga. K-means clustering is a MATLAB® code package available in MATLAB version R2006a onwards.

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

## Acknowledgements

This data used in this study were collected during Natural Environment Research Council (NERC) grant NE/K010743/1 (VoiLA, Volatile Recycling in the Lesser Antilles) with seabed instruments from the UK Ocean-Bottom Instrumentation Facility (OBIF)[86] and the German Instrument Pool for Amphibian Seismology (DEPAS), hosted by the Alfred Wegener Institute Bremerhaven. We acknowledge Halliburton for providing access to SeisSpace®/ProMax® software via a grant to Imperial College London. Thanks go to R. G. Davy for conducting hodogram analysis, to Linfeng Li, Chao Sun, and Mingrui Li for the discussion about K-means cluster analysis, DEM analysis and water content calculations. Lianjun Li is supported by Imperial College London President's PhD Scholarship Scheme and Special Youth Fund for Science and Technology of China National Petroleum Corporation (Grant No. 2024DQ03015). All figures were prepared using the Generic Mapping Tools[87] and Adobe Illustrator®.

## Author contributions

L.L. conducted the OBS wide-angle seismic data analysis, downward continuation, travel time data picking, tomographic modelling and wrote the initial draft. J.C. is the PI of the project. L.L. and J.C. conducted the gravity modelling, DEM analysis and water content calculations. T.H. and S.G. are the co-PIs of the project. T.H. conducted initial onboard processing of the data. L.L. and S.G. conducted the K-means cluster

analysis. All authors contributed to the interpretation and presentation of the results, manuscript writing, reviewing and editing.

## Competing interests

The authors declare no competing interests.
