## [Transparent Peer Review file · Nature Communications]

Estimating excess bound water content due to serpentinisation in mature slow-spreading oceanic crust using Vp/Vs

Corresponding Author: Ms Lianjun Li

Version 0:

Reviewer comments:

Reviewer #1

(Remarks to the Author)

Li and coauthors present new constraints on the hydration state of the Atlantic lithosphere outboard of the Lesser Antilles trench from P-wave, S-wave and Vp/Vs constraints. As the authors write, many constraints on hydration of oceanic crust and upper mantle come from P-wave velocity models, but other factors besides serpentinization can influence Vp, rendering those estimates uncertain. Estimating hydration is especially challenging in slow-spreading crust, which has significant heterogeneity. This paper presents relatively high-resolution models of Vp, Vs and (from those) Vp/Vs that provide much better constraints on both the distribution and extent of hydration in the lithosphere. The profile is relatively long and crosses both relatively magma-rich and magma-poor segments. To my knowledge, this is now the best model we have of Atlantic crustal/upper mantle structure and hydration, and it has important implications for controls on hydration in slow-spreading oceanic lithosphere and the subduction zone water cycle. The analysis, interpretation, writing and illustration are all excellent. I strongly recommend publication and only have minor comments, questions and suggestions described below. My lengthy review reflects my interest in this work.

-The interpretation of variations in velocity structure in terms of magma-poor and magma rich segments appears convincing, but it would be even stronger if there was other evidence or rationale to support it. Are there any known changes in spreading rate between segments or other constraints elsewhere in these segments that support the classification presented here?

-The authors interpret variations in upper mantle velocity in terms of composition (depletion/enrichment). What impact would upper mantle anisotropy have on upper mantle Vp, Vs and Vp/Vs? This profile is orthogonal to the spreading direction, and previous studies elsewhere in the Atlantic observe some degree of azimuthal anisotropy in the upper mantle (albeit weaker than in fast spreading Pacific lithosphere) (e.g., Gaherty et al., EPSL, 2004)

-I think it would be good to expand the discussion of the influence of the Barracuda Ridge on velocities near OCC-3 and OCC-4. Studies elsewhere have shown that there can be significant water stored in the upper crust of volcanic ridges and seamounts (e.g., Chesley et al., Nature, 2021) – could the high Vp/Vs and low Vs and Vp around OCC-4 be explained by a porous, water rich upper volcanic crust of Barracuda Ridge? Also, is the crust in this area thicker than it would be otherwise because of proximity to Barracuda Ridge?

-Lines 335-340 and abstract. Enhanced hydration around fracture zones is not just assumed or a common view – it has been demonstrated with observations in several regions (e.g., Roland et al., JGR, 2012). I think it would be more interesting to frame the discussion of this topic to explore why hydration does not appear concentrated along fracture zones in this area in contrast to other areas (instead of saying that these results suggest hydration along fracture zones is not as important as previously thought). Perhaps the contrast in hydration between fracture zones and adjacent segments is more pronounced in fast-spreading lithosphere (where many studies of FZ hydration have been done) than slow spreading lithosphere? In the abstract (Line 31), I do not think that previous studies assume hydration is limited to FZ in slow spreading crust, where OCC and thin, hydrated crust/upper mantle have long been observed, so I suggest modifying that.

-One of the most interesting observations of the model is the “patchiness” of crustal hydration. In addition to contributing to

the total water delivered into the Lesser Antilles subduction zone, I think that the irregular distribution could also be important for many processes at depth in the subduction zone. For example, numerical models that include either uniform hydration of variable hydration (but same total water) show different dehydration patterns that could be important for deformation and magmatism (e.g., Wada et al., EPSL, 2012). I think it would be interesting to explore these implications in the Discussion.

Other minor comments/questions:

-The title only mentions S-waves, but it is the combination of high-quality Vp and Vs models (and Vp/Vs) that allows the authors to better quantify crustal structure and hydration. Consider changing the title to "...oceanic crust using Vp/Vs"?

-Lines 70-72. A reference should be provided for this statement.

-Line 119. This sentence is confusing, contradictory as written. "a single velocity-gradient structure with velocity inversions". If there is a velocity inversion, then one could not describe it as having a single velocity gradient?

-Lines 211-213. Is a difference in misfit of only 2 mGal significant? This seems like a very small improvement. Additionally, one of these sentences is confusing as written. "...an overall misfit of 8 mGal, but locally the misfit exceeds ± 3 mGal".

-Line 252. I was surprised to read that the upper 2 km of the crust have "negligible hydration". This seems inconsistent with many other studies showing extensive hydration of the upper oceanic crust from hydrothermal circulation and off-axis fluid flow until hydration products and/or sediments prevent it, including from a recent drilling and geophysical transect in the southern Atlantic (e.g., Kardell et al., JGR, 2019).

-Line 282. Suggest editing to "This results in the highest..."

-Line 545. I thought the use of the K-means cluster analysis for grouping areas with different velocity characteristics was novel. Has this type of analysis been used for classifying velocities? If so, it would be good to cite those studies. If not, I think the Methods (or main paper) could emphasize the novelty.

-Suppl Fig 3 – It would be useful if this figure could be made larger and higher resolution so that it is easier to see the data and predicted/observed travel times. I cannot see some of the symbols in the key in the panels.

-Several of the supplemental figures show a basement interface based on MCS data. Were MCS data used to constrain the basement interface, or did it only come from DC OBS data? If MCS data were used to constrain the models, it would be good to include a plot of the MCS data in the supplement and mention how the MCS data were used in the Methods.

Thank you for the opportunity to review this interesting paper.

Donna Shillington

Reviewer #2

(Remarks to the Author)

Li et al. present a detailed study of oceanic crust created in a slow-spreading environment. The main importance of this work is the measurement of both P and S velocities (Vp, Vs) which are critical to interpreting lithology. Estimation of Vs from active-source experiments is uncommon because it requires identifying P-to-S converted phases (water column source cannot produce a direct S-wave). Using their 2D velocity model, they investigate controls on the hydration state of the crust. They find that serpentinized oceanic core complexes may contribute significantly to subduction zone water budgets. I find the study well-presented and it would be of interest to the Nature Communications audience. However, I have a few concerns regarding the lithologic interpretations and model resolution that require moderate revision before publication.

General Comments:

1. Hydration Estimates

A major result of this work is that oceanic core complexes (OCCs) are highly serpentinized and, thus, contribute significantly to subduction zone water budgets. However, the water content calculations within these zones is likely an over-estimate because the effect of fluid-filled fractures on P and S velocities is not considered.

1a. While the total volume occupied by fluid-filled cracks is small, they can significantly reduce seismic velocities and alter Vp/Vs ratios (e.g., Kuster & Toksöz; 1974). Appealing to fractures, one could achieve similar velocity reductions with much less water storage in OCCs. Considering that the inferred most hydrated zones are relatively shallow (2-3 km depth; Fig. 5b) and occupy deformed regions of seafloor, this mechanism is likely relevant but not thoroughly considered in the present study. I would like to see a second lower-bound estimate of hydration by assuming all water is stored within fractures.

1b. Mechanisms for anomalous velocities beyond serpentinisation are also suggested by Fig. 5a. This plot serves as the basis for the lithologic/hydration interpretations. However, the Vp and Vp/Vs values for the inferred hydrated ultramafic model points have a very different trend from that predicted by Carlson and Miller or Horen et al. (bias toward high Vp/Vs at

lower Vp relative to the alteration curves). In contrast, the drilled ultramafic samples (magenta points), despite some scatter, do follow the plotted serpentinisation trend. This point warrants some discussion in the text because the velocity trends appear to argue for a different ultramafic hydration mechanism than the one invoked by the authors. It may be informative to compare Vp, Vp/Vs trends resulting from pore/fracture water in Fig. 5 (or elsewhere).

2. Model Resolution

2a. I think an important point regarding the resolution is how picking uncertainty effects the recovery of Vp/Vs ratios. The picking error for Sg (~177 ms) is ~5x larger than for Pg (~32 ms). Therefore, I would expect the amplitude recovery of Vs anomalies to be poorer than Vp anomalies; the resulting Vp/Vs ratios would then be biased. Do the synthetic data used in the resolution tests (Supplementary Fig. 15) include this data quality discrepancy (e.g., adding gaussian noise with different standard deviations for P and S travel-times)?

2b. Considering that the interpretation of Vp/Vs ratios is a key aspect of this work, I would like to see resolution tests specifically targeting Vp/Vs (and include differences in data quality mentioned above). It is difficult to understand from Supplementary Fig. 15 alone how well the target model Vp/Vs ratios are imaged. It also seems that the inversion over-estimates anomaly amplitudes considering many recovered perturbations saturate the color scale.

Specific Comments:

Line 25-27: I suggest revising lines 25-27 to emphasise the lack of shear velocity constraints on oceanic crustal structure as this is the real novelty of the study.

Line 24: Here, and several other places, the term 'slow-spread crust' is used but I believe 'slow-spreading crust' is the more common phrase. Consider revising.

Line 52: Can you provide a reference for Penrose stratigraphy?

Line 245-246: Could you better explain why these pink samples in particular are difficult to assign a composition? Specifically, it is not clear why the boundaries of the pink samples doesn't extend to lower Vp or above the Vp/Vs = 1.9 line (presumably samples near this boundary, above or below, are difficult to identify).

Line 282-283: Considering these are relatively shallow anomalies in a rather deformed setting, to what extent could these velocity reductions and Vp/Vs anomalies be due to fracturing as opposed to serpentinisation?

Line 433: Can you clarify here that PSP arrivals are waves that travel through the sediments at source and receiver sides as P but through the crust/mantle as S?

Line 522; 536: Please clarify that these velocity uncertainty estimates are due to sensitivity to the starting model rather than the true error in the recovered parameter values.

Line 593-594: Please clarify how you arrived at a density anomaly of -0.15 g/cc for the gravity modelling.

Line 602-604: As written, it sounds like you've established your own lithologic boundaries based on body wave velocities from field/laboratory data (Fig. S29). However, you do not provide sufficient details on how these boundaries were determined. While you cite Grevemeyer et al. in regards to this, you don't explain how or why you've modified the lithologic boundaries defined in the Grevemeyer study. Please clarify.

Supplementary Figure S29: It is a bit unclear from the caption or text what exactly the narrow light blue and yellow linear bands are showing. I'm guessing that the light-blue band is the peridotite-serpentinite trend along which the degree of serpentinisation is increasing. However, the fraction of serpentinite (or water content) along this curve is not indicated. Presumably the light-yellow band is the gabbro-dolerite trend along which the fraction of dolerite increases? Can the fraction of dolerite be labeled on this curve? Please clarify this in the caption/figure.

Version 1:

Reviewer comments:

Reviewer #1

(Remarks to the Author)

The authors have undertaken a thorough revision of the manuscript based on the reviewer comments. All of my questions and concerns have been addressed, and it appears that the authors have included additional calculations to address one of the central comments of the other reviewer, which have also improved the manuscript.

As described in my previous review, I think this manuscript is very well written and illustrated, and it presents an important new result with implications for processes at slow-spreading mid-ocean ridges and subduction zones. I recommend publication.

Reviewer #2

(Remarks to the Author)

This is my second review of the Li et al. manuscript. The authors have thoughtfully addressed all my comments and I look forward to seeing the work published in Nature Communications.

Response to Reviewers' comments on Nature Communications manuscript NCOMMS-24-69268

“Estimating excess bound water content due to serpentinitisation in mature slow-spread oceanic crust using S-waves” by Li et al.

We thank the reviewers for their insightful comments which we believe has significantly improved the clarity and robustness of the manuscript.

Below reviewers' comments are in *blue italics* and our responses in **black**. During the revision we also followed the request of the Editor and made some edits to meet the Nature Communications format requirements. As a result, a section about the oceanic core complexes that was in the Discussion has moved to the Results. Below we refer to line/page/paragraph numbers in the **revised version**. Please note that the separate PDF version of Supplementary Figures has higher resolution compared to the PNG file inserted in the merged supplementary file.

Reviewer 1: Donna Shillington

General Comments:

Li and coauthors present new constraints on the hydration state of the Atlantic lithosphere outboard of the Lesser Antilles trench from P-wave, S-wave and Vp/Vs constraints. As the authors write, many constraints on hydration of oceanic crust and upper mantle come from P-wave velocity models, but other factors besides serpentinitization can influence Vp, rendering those estimates uncertain. Estimating hydration is especially challenging in slow-spreading crust, which has significant heterogeneity. This paper presents relatively high-resolution models of Vp, Vs and (from those) Vp/Vs that provide much better constraints on both the distribution and extent of hydration in the lithosphere. The profile is relatively long and crosses both relatively magma-rich and magma-poor segments. To my knowledge, this is now the best model we have of Atlantic crustal/upper mantle structure and hydration, and it has important implications for controls on hydration in slow-spreading oceanic lithosphere and the subduction zone water cycle. The analysis, interpretation, writing and illustration are all excellent. I strongly recommend publication and only have minor comments, questions and suggestions described below. My lengthy review reflects my interest in this work.

In making our revision we have borne in mind this very positive review. We believe that the edits made during the revision have further improved the manuscript.

1.1 Spreading rate estimates

The interpretation of variations in velocity structure in terms of magma-poor and magma rich segments appears convincing, but it would be even stronger if there was other evidence or rationale to support it. Are there any known changes in spreading rate between segments or other constraints elsewhere in these segments that support the classification presented here?

Unfortunately, at these slow spreading rates, the magnetic anomalies are not separated enough to date the seafloor at the resolution that would be needed to recognise local spreading rate variations. There is of course also an inherent asymmetry to magma-poor spreading which means that half spreading rate may be a poor proxy for magma budget in any case. Numerical modelling by Gerya's group (Liu et al., 2025; Liu et al., 2022) shows that magma-robust and

magma-poor segments emerge self-consistently in a setting of slow-ultraslow spreading, even if rates are constant along the ridge. In this case, variations in spreading rate, fault weakening (e.g. by variations in fluid infiltration) and mantle temperature in time can explain shifting patterns of magma-robust and magma-poor in time (i.e. in ridge-perpendicular direction). We have added this point to the discussion (line 371).

1.2 Effect of mantle anisotropy

The authors interpret variations in upper mantle velocity in terms of composition (depletion/enrichment). What impact would upper mantle anisotropy have on upper mantle Vp, Vs and Vp/Vs? This profile is orthogonal to the spreading direction, and previous studies elsewhere in the Atlantic observe some degree of azimuthal anisotropy in the upper mantle (albeit weaker than in fast spreading Pacific lithosphere) (e.g., Gaherty et al., EPSL, 2004)

We omitted to say that we find very similar mantle velocities on our Line 1 profile (Fig. 1a; Allen et al., 2022) which is orthogonal to Line 2/3 presented here. We therefore did not consider anisotropy a major factor which is why we speculated that depletion/enrichment could be underlying cause of our observations. We have corrected this omission (line 331).

1.3 Influence of the Barracuda Ridge on velocity structure

I think it would be good to expand the discussion of the influence of the Barracuda Ridge on velocities near OCC-3 and OCC-4. Studies elsewhere have shown that there can be significant water stored in the upper crust of volcanic ridges and seamounts (e.g., Chesley et al., Nature, 2021) – could the high Vp/Vs and low Vs and Vp around OCC-4 be explained by a porous, water rich upper volcanic crust of Barracuda Ridge? Also, is the crust in this area thicker than it would be otherwise because of proximity to Barracuda Ridge?

This comment prompted us to recognise that we should have provided a clearer introduction to the Barracuda Ridge. Whilst it may have originally formed as a transverse ridge, in a similar way to those in the nearby Vema, St Paul and Romanche fracture zones (Bonatti, 1978; Bonatti et al., 2005), it appears to have been uplifted by later tectonic compression across the North American-South American plate boundaries (Bouysse and Westercamp, 1990; Pichot et al., 2012; Roest and Collette, 1986; Stein et al., 1982). In this way it is more similar to Gorrington Bank in the eastern Atlantic at the African-Eurasian plate boundary. We have added a sentence in the Introduction section and (line 91-95) to explain this. The effect of fracturing on seismic velocity is now also included in response to comments from Rev 2 (see below).

1.4 Fracture zone hydration concepts

Lines 335-340 and abstract. Enhanced hydration around fracture zones is not just assumed or a common view – it has been demonstrated with observations in several regions (e.g., Roland et al., JGR, 2012). I think it would be more interesting to frame the discussion of this topic to explore why hydration does not appear concentrated along fracture zones in this area in contrast to other areas (instead of saying that these results suggest hydration along fracture zones is not as important as previously thought). Perhaps the contrast in hydration between fracture zones and adjacent segments is more pronounced in fast-spreading lithosphere (where many studies of FZ hydration have been done) than slow spreading lithosphere? In the abstract (Line 31), I do not think that previous studies assume hydration is limited to FZ in slow spreading crust, where OCC and thin, hydrated crust/upper mantle have long been observed, so I suggest modifying that.

This comment helped us clarify our presentation and we bore this comment in mind when making the revisions. In particular, we have rewritten the second paragraph of the discussion starting with “*Many previous studies have demonstrated that fracture zone...*”(line 360). We also removed the phrase “*is often assumed*” and “*as traditionally thought*” to present the results in a more neutral and open manner.

1.5 Expand discussion on the effect of “patchiness” of crustal hydration

One of the most interesting observations of the model is the “patchiness” of crustal hydration. In addition to contributing to the total water delivered into the Lesser Antilles subduction zone, I think that the irregular distribution could also be important for many processes at depth in the subduction zone. For example, numerical models that include either uniform hydration of variable hydration (but same total water) show different dehydration patterns that could be important for deformation and magmatism (e.g., Wada et al., EPSL, 2012). I think it would be interesting to explore these implications in the Discussion.

Yes, this is a good point. We have reorganised the final paragraph of the Discussion to give more prominence to the “patchiness” result and its broader implications.

Minor Comments:

- *The title only mentions S-waves, but it is the combination of high-quality Vp and Vs models (and Vp/Vs) that allows the authors to better quantify crustal structure and hydration. Consider changing the title to “...oceanic crust using Vp/Vs”?*

Good point. We have changed the title to “using Vp/Vs” (and so still remain within the required number of words for the title).

- *Lines 70-72. A reference should be provided for this statement.*

Done (Merdith et al., 2019). As a result of this change, we also replaced the first reference with a more recent overview paper on the question of mantle water budgets over geological time (Rüpke and Gaillard, 2024)

- *Line 119. This sentence is confusing, contradictory as written. “a single velocity-gradient structure with velocity inversions”. If there is a velocity inversion, then one could not describe it as having a single velocity gradient?*

Good point. We have rewritten this sentence as “a single velocity-gradient structure with occasional velocity inversions identified as OCCs superimposed” (line 112).

- *Lines 211-213. Is a difference in misfit of only 2 mGal significant? This seems like a very small improvement. Additionally, one of these sentences is confusing as written. “...an overall misfit of 8 mGal, but locally the misfit exceeds ± 3 mGal”.*

Good point. While 2 mGal indeed does not seem very significant, this value is averaged across the entire line and so includes the sections between OCCs that remain unchanged in the two models. These unchanged sections account for approximately 80% of the line length (the parts shown with red and blue striped line in Fig 4d). We intended to express that “locally the misfit exceeds the assigned uncertainty, i.e., ± 3 mGal (the parts indicated with yellow arrows in the

figure).” After adding negative density anomalies, the areas with the largest misfit are significantly reduced. We have revised both the main text and the figure caption accordingly.

- *Line 252. I was surprised to read that the upper 2 km of the crust have “negligible hydration”. This seems inconsistent with many other studies showing extensive hydration of the upper oceanic crust from hydrothermal circulation and off-axis fluid flow until hydration products and/or sediments prevent it, including from a recent drilling and geophysical transect in the southern Atlantic (e.g., Kardell et al., JGR, 2019).*

We thank the reviewer for this comment. Line 252 in the original manuscript actually used the phrase “negligible excess hydration” which was intended to mean little or no water content above that expected for mature Penrose-style oceanic crust. This reviewer’s comment suggests that our use of “excess” did not come across clearly enough and this is an important point for the overall understanding of the manuscript. We have added the sentence “*In other words, according to our classification, the hydration levels of these segments do not surpass what would be expected for Penrose-style oceanic crust of this age*” to make our meaning clearer. We made minor revisions elsewhere in the manuscript to reinforce our intended use of the term “excess hydration” (line 324).

- *Line 282. Suggest editing to “This results in the highest...”*

Done (line 244).

- *Line 545. I thought the use of the K-means cluster analysis for grouping areas with different velocity characteristics was novel. Has this type of analysis been used for classifying velocities? If so, it would be good to cite those studies. If not, I think the Methods (or main paper) could emphasize the novelty.*

To our knowledge this has not been conducted before. We have added a sentence to the methods section stating this as suggested (line 574).

- *Suppl Fig 3 – It would be useful if this figure could be made larger and higher resolution so that it is easier to see the data and predicted/observed travel times. I cannot see some of the symbols in the key in the panels.*

Done. We also did a general review of all font sizes etc. Please note that the separate PDF version of Supplementary Fig.3 has higher resolution compared to the PNG file inserted in the merged supplementary file.

- *Several of the supplemental figures show a basement interface based on MCS data. Were MCS data used to constrain the basement interface, or did it only come from DC OBS data? If MCS data were used to constrain the models, it would be good to include a plot of the MCS data in the supplement and mention how the MCS data were used in the Methods.*

Done. We have added the co-incident, pre-stack time migrated MCS seismic profile as the top panel in Supplementary Fig. 7 and revised the figure caption. The initial top basement horizon included in the *P*-wave sedimentary starting model (Supplementary Fig. 7b) was picked from this MCS profile and then depth converted using the initial sediment model (Supplementary Fig.

7b). During the inversion of the sedimentary Vp model, the top basement was further constrained using PbP phases picked from DC OBS data. The comparison between the initial and final inverted top basement is shown in Supplementary Fig. 7d (black dashed line and red line, respectively). More details about inverting the sedimentary Vp model and top basement using DC OBS data can be found in Li et al. (2024).

Reviewer 2: Anonymous

General Comments:

Li et al. present a detailed study of oceanic crust created in a slow-spreading environment. The main importance of this work is the measurement of both P and S velocities (Vp, Vs) which are critical to interpreting lithology. Estimation of Vs from active-source experiments is uncommon because it requires identifying P-to-S converted phases (water column source cannot produce a direct S-wave). Using their 2D velocity model, they investigate controls on the hydration state of the crust. They find that serpentinised oceanic core complexes may contribute significantly to subduction zone water budgets. I find the study well-presented and it would be of interest to the Nature Communications audience. However, I have a few concerns regarding the lithologic interpretations and model resolution that require moderate revision before publication.

We are grateful for the reviewer's acknowledgement of this work's novelty and substantive contributions. The commentary on the analysis fluid-filled fractures and model resolution has helped us to strengthen our interpretation in a rather complete manner.

2.1. Hydration estimates

A major result of this work is that oceanic core complexes (OCCs) are highly serpentinized and, thus, contribute significantly to subduction zone water budgets. However, the water content calculations within these zones is likely an over-estimate because the effect of fluid-filled fractures on P and S velocities is not considered. While the total volume occupied by fluid-filled cracks is small, they can significantly reduce seismic velocities and alter Vp/Vs ratios (e.g., Kuster & Toksöz; 1974). Appealing to fractures, one could achieve similar velocity reductions with much less water storage in OCCs.

We thank the reviewer for this comment and acknowledge that we should have addressed the effect of fracturing on Vp/Vs ratios in the original manuscript. We have now incorporated a discussion of this effect as outlined below. We recognise that by adopting the serpentinite endmember as our preferred interpretation, our excess water content estimate is indeed an upper bound. We have taken this point on-board and revised the manuscript accordingly.

We now include Differential Effective Medium (DEM) calculations for fractured media following the methods outlined in (Mukerji et al., 1995; Norris, 1985; Taylor and Singh, 2002). We show an example of the predictions in Figure 6b, and additional documentation in Supplementary Table 3 and Supplementary Figs. 30—32. In the main text, we modified accordingly the subsection of the Results section: “Lithology, serpentinisation, water-filled fractures and physical properties”, (line 268). We added a new Methods section: “Relationship between fractures and Vp/Vs” (starting line 638) and included alternate models in the section titled “Water content calculations for anomalous crust” (starting line 652).

2.1a Mechanisms for giving anomalous velocity

Mechanisms for anomalous velocities beyond serpentinisation are also suggested by Fig. 5a. This plot serves as the basis for the lithologic/hydration interpretations. However, the V_p and V_p/V_s values for the inferred hydrated ultramafic model points have a very different trend from that predicted by Carlson and Miller or Horen et al. (bias toward high V_p/V_s at lower V_p relative to the alteration curves). In contrast, the drilled ultramafic samples (magenta points), despite some scatter, do follow the plotted serpentinisation trend. This point warrants some discussion in the text because the velocity trends appear to argue for a different ultramafic hydration mechanism than the one invoked by the authors. It may be informative to compare V_p , V_p/V_s trends resulting from pore/fracture water in Fig. 5 (or elsewhere).

We test dry gabbro and peridotite as the rock matrix materials (to match the dominant rock types of OCCs from IODP drilling (Blackman et al., 2019; Canales et al., 2008)) and water-filled cracks with various aspect ratios. As the reviewer correctly states, fractures can explain the spread of $V_p/V_s > 1.9$ values we obtain from our seismic model results about the predicted ultramafic-serpentinisation trend. Specifically, very thin cracks (with aspect ratios 0.0001) using a peridotite rock matrix are required to explain our seismic model data samples with V_p above the serpentinite trend, and thicker cracks (with aspect ratios between 0.005 and 0.013) are needed to match data samples with V_p below the serpentinite trend (Fig. 6b).

However, in the original submission we did not make good use of our gravity modelling results. In the revised Fig.6 we do this more explicitly as a means of discriminating between serpentinization and fracture mechanisms. For the anomalous $V_p/V_s > 1.9$ regions, the seismic data samples indicate a range of fracture porosities of between 0.02 and 5.69 % (with the bulk of the data samples at the low end of this range). However, the density reduction that would occur with such a fracture population is insufficient to account for our gravity model results, which require a drop in these regions of 0.5 g/cm^3 . This reduction is easily achieved by serpentinisation at the levels indicated by our V_p/V_s observations. For this reason, in the final part of the Results section we focus on the serpentinisation model, but alternates are given in the Supplementary Fig. 31-32 (paragraphs starting line 664 and 674).

2.1b Implications for excess water estimates

Considering that the inferred most hydrated zones are relatively shallow (2-3 km depth; Fig. 5b) and occupy deformed regions of seafloor, this mechanism is likely relevant but not thoroughly considered in the present study. I would like to see a second lower-bound estimate of hydration by assuming all water is stored within fractures.

We now set up and evaluate three hydration models in the final section of the Methods as follows: (i) a model in which $V_p/V_s > 1.9$ is interpreted as hydrated ultramafic rocks (as previously but we now exclude the pink-coloured data samples from the calculation, Fig. 6a and Fig. 7b) (ii) a model in which $V_p/V_s > 1.9$ is interpreted as fractured ultramafic/mafic rocks (Fig. 6b, Supplementary Figs. 31—32) and (iii) a hybrid model (Supplementary Figs. 31—32) in which the data samples with $V_p/V_s > 1.9$ that lie on the serpentine hydration trend are interpreted as hydrated ultramafic rocks and those that lie off this trend are interpreted as fractured ultramafic rocks. Note that we choose only fractured ultramafic rock as an example in the hybrid model as the fracture-water template of a peridotite matrix can cover all velocity points with $V_p/V_s > 1.9$ and also the overall fracture water content estimated based on different rock matrices is similar (Supplementary Fig. 32b, c) which therefore has little effect on the hydration estimation in a hybrid mode.

Model 1 is our strongly preferred interpretation, based the geometry and internal structure of the anomalous Vp/Vs bodies, their proximity to elevated or exposed basement and their gravity signature. Model 2 is ruled out from density reduction based on Fig. 6c. This model results in the uppermost and central core of the OCCs regions that are dryer than the surrounding area. We believe it is unlikely to maintain water within fractured peridotite without alteration in a 65 Ma lithosphere and given the reactivity of seawater-olivine, and the widespread occurrence of serpentinite in the oceans that model 2 is not a realistic model (e.g., Bach and Fruh-Green, 2010; Klein et al., 2015; Mevel, 2003). Nonetheless, it may serve as a useful end-member as we think reality is much closer to pure alteration (serpentinisation), followed by a combination of alteration and fracturing and then the pure fracturing model. We therefore use Model 3 to give our minimum bound excess water content estimate. We have added a related description for these models to the main text in the Results section “*Relationships between lithology, serpentinisation, water-filled fractures and physical properties*”, “*Preferred lithological model and excess hydration due to serpentinisation*” and in the Methods section “*Relationship between lithology, serpentinisation and Vp/Vs*”, “*Relationship between fractures and Vp/Vs*” and “*Water content calculations for anomalous crust*”.

2.2 Model Resolution

2.2a Vp and Vs pick uncertainties and recovery

I think an important point regarding the resolution is how picking uncertainty effects the recovery of Vp/Vs ratios. The picking error for Sg (~177 ms) is ~5x larger than for Pg (~32 ms). Therefore, I would expect the amplitude recovery of Vs anomalies to be poorer than Vp anomalies; the resulting Vp/Vs ratios would then be biased. Do the synthetic data used in the resolution tests (Supplementary Fig. 15) include this data quality discrepancy (e.g., adding gaussian noise with different standard deviations for P and S travel-times)?

We thank the reviewer for this comment, as indeed we had not included the difference in picking uncertainty when generating the results for checkerboard tests. We have now done this and updated Supplementary Fig.15 and the caption. We have modified lines 529-534 in the Methods section (“*Quality analysis and checks for final P- and S-wave models*”) as follows “Checkerboard tests performed with a range of velocity anomaly sizes at an amplitude of $\pm 5\%$ and with a gaussian noise level (standard deviation of 10 ms, 50ms, and 100ms) scaled by the picking uncertainty for P- and S-waves, reveal a resolution for Vp of 5×1 to 5×1.5 km in the upper crust, 5×2.5 km in the lower crust and 10×5 km in the upper mantle, and for Vs of 5×2.5 km to 10×5 km in the crust and 15×5 km in the upper mantle (Supplementary Fig. 15)” and updated the resolution box in Fig.2 and its caption accordingly. Please note that the separate PDF version of Supplementary Fig.15 has higher resolution compared to the PNG file inserted in the merged supplementary file.

The results show that the amplitude recovery is indeed poorer for Vs as the reviewer predicted. Increasing the gaussian noise level with different standard deviations added to the synthetic picks affects the recovery of small-scale sinusoidal velocity anomalies (e.g., 5×1 km and 5×1.5 km) while the differences are only subtle for the recovery of large-scale anomalies (e.g., 10×5 km and 15×5 km).

2.2b Resolution of Vp/Vs

Considering that the interpretation of Vp/Vs ratios is a key aspect of this work, I would like to see resolution tests specifically targeting Vp/Vs (and include differences in data quality mentioned above). It is difficult to understand from Supplementary Fig. 15 alone how well the target model Vp/Vs ratios are imaged. It also seems that the inversion over-estimates anomaly amplitudes considering many recovered perturbations saturate the color scale.

We thank the reviewer for this comment as indeed the interpretation of Vp/Vs ratios is the basis of this work. As Vp/Vs model is an indirect product out of the direct tomographic inversion of Vp and Vs travel time data, it would be difficult to conduct a similar checkerboard routine, as we do not invert Vp/Vs model from travel time data. However, the resolution of Vp/Vs depends on the lower resolution of Vs as the finer details of Vp are “downsampled” to match the Vs when computing Vp/Vs. In Supplementary Fig. 15, we added black dashed lines to help compare the size of identified Vp/Vs anomalous features in Fig. 2 to the size of the checkers. Vs in the crustal column has a resolution in general at 10×5 km (horizontal \times vertical), finer than the size of identified high/low Vp/Vs features at OCC-2 ($x=225-270$ km), OCC-3 ($x=55-75$ km), OCC-4 ($x=75-110$ km) and OCC-1 (VI-3) ($x=210-225$ km). Especially, although the amplitude is recovered relatively poorly, the Vs resolution achieved at 5×2.5 km at OCC-5 ($x\sim 300$ km) and VI-2 ($x\sim 135$ km), which is also finer than the size of these Vp/Vs features. Therefore, we concluded that the interpreted Vp/Vs features are reliable for the analysis.

Specific Comments:

- *Line 25-27: I suggest revising lines 25-27 to emphasise the lack of shear velocity constraints on oceanic crustal structure as this is the real novelty of the study.*

Done

- *Line 24: Here, and several other places, the term 'slow-spread crust' is used but I believe 'slow-spreading crust' is the more common phrase. Consider revising.*

Done

- *Line 52: Can you provide a reference for Penrose stratigraphy?*

Done (Anonymous, 1972)

- *Line 245-246: Could you better explain why these pink samples in particular are difficult to assign a composition? Specifically, it is not clear why the boundaries of the pink samples doesn't extend to lower Vp or above the Vp/Vs = 1.9 line (presumably samples near this boundary, above or below, are difficult to identify).*

In our revised manuscript, we strictly use Vp/Vs=1.9 to differentiate serpentinised peridotite from others so we now exclude the previous pink-coloured data samples from the calculation to avoid ambiguity. The modification is applied to Figs. 6-7.

In the previous version, the serpentinisation trend lines (blue lines in Supplementary Fig. 29) and lithology (dolerite/gabbro, yellow lines in Supplementary Fig. 29) overlap between Vp=6.5-7.5 km/s and Vp/Vs=1.85-1.9 (the region of the pink samples). In our original analysis, we assigned this region to be partially serpentinised rock based on its spatial position within our seismic velocity model (pink areas in original Fig. 5). However, based on the reviewer's

comment, we have now removed these velocity points from our model (i) estimate of excess water content described above. We note, however, that due to their small area and relatively low water contents (3-6 wt% H₂O), removing these data points only reduces our maximum bound water content estimate from 4.5 wt% to 4 wt%. Please note that in the original manuscript, we made a mistake in calculating the average bound water content over the whole crustal column. The average was not calculated through the whole crustal column but mistakenly over the ranges where bound water content is above the minimum assigned value. So, the number was calculated as large as 6.6 wt%. We have modified this according to the abstract and the Discussion section.

- *Line 282-283: Considering these are relatively shallow anomalies in a rather deformed setting, to what extent could these velocity reductions and Vp/Vs anomalies be due to fracturing as opposed to serpentinisation?*

See discussion of point 2.1 above.

- *Line 433: Can you clarify here that PSP arrivals are waves that travel through the sediments at source and receiver sides as P but through the crust/mantle as S?*

Yes, we have done this and added more clarification about the point of conversion throughout this section of the methods. This is now modified as “This delay was subtracted from Sg/Sn/SmS arrivals to produce “effective PSP arrivals”(Eccles et al., 2009) i.e. as if they had travelled through the sediments at both source and receiver sides as P-waves but through the crust/mantle as S-waves in order to give symmetry for tomographic inversion using Vp in the sediments (Supplementary Fig. 4).”

- *Line 593-594: Please clarify how you arrived at a density anomaly of -0.15 g/cc for the gravity modelling.*

We have rewritten the text and added two references to explain how we obtained this value (see Methods section “Gravity modelling”). In the revised manuscript, the gravity results are used more explicitly to assess whether the upper crustal bodies with the anomalous Vp/Vs ratios have densities that would be expected if they contained significant quantities of serpentinite as opposed to fractured ultramafic/mafic rocks. See discussion of point 2.1 above.

- *Line 602-604: As written, it sounds like you've established your own lithologic boundaries based on body wave velocities from field/laboratory data (Fig. S29). However, you do not provide sufficient details on how these boundaries were determined. While you cite Grevemeyer et al. in regards to this, you don't explain how or why you've modified the lithologic boundaries defined in the Grevemeyer study. Please clarify.*

The reviewer makes a fair point. We have re-written this section of the methods to be more direct. Changes are made to Methods section “Relationships between lithology, serpentinisation and Vp/Vs” (starting line 624).

- *Supplementary Figure S29: It is a bit unclear from the caption or text what exactly the narrow light blue and yellow linear bands are showing. I'm guessing that the light-blue band is the*

peridotite-serpentinite trend along which the degree of serpentinisation is increasing. However, the fraction of serpentinite (or water content) along this curve is not indicated. Presumably the light-yellow band is the gabbro-dolerite trend along which the fraction of dolerite increases? Can the fraction of dolerite be labeled on this curve? Please clarify this in the caption/figure.

We have rewritten the figure caption to explain the rock trends more clearly. We have also replotted the figure to make the numbers indicating the fraction of serpentinite (or water content) along the light-blue band more visible (previously, some of the values were obscured which may have led to the reviewer missing them). We prefer not to include the fraction of dolerite as the figure is already quite complex, and the main aim of the manuscript is to identify and interpret regions that depart from the Penrose stratigraphy.

References

- Allen, R.W., Collier, J.S., Henstock, T.J., 2022. The Role of Crustal Accretion Variations in Determining Slab Hydration at an Atlantic Subduction Zone. *Journal of Geophysical Research: Solid Earth* 127, e2022JB024349.
- Anonymous, 1972. Penrose field conference on ophiolites. *Geotimes* 17, 24–25.
- Bach, W., Fruh-Green, G.L., 2010. Alteration of the Oceanic Lithosphere and Implications for Seafloor Processes. *Elements* 6, 173-178.
- Blackman, D.K., Abe, N., Carlson, R.L., Guerin, G., Ildefonse, B., Kumpf, A., 2019. Seismic properties of gabbroic sections in oceanic core complexes: constraints from seafloor drilling. *Marine Geophysical Research* 40, 557-569.
- Bonatti, E., 1978. Vertical tectonism in oceanic fracture zones. *Earth and Planetary Science Letters* 37, 369-379.
- Bonatti, E., Brunelli, D., Buck, W., Cipriani, A., Fabretti, P., Ferrante, V., Gasperini, L., Ligi, M., 2005. Flexural uplift of a lithospheric slab near the Vema transform (Central Atlantic): Timing and mechanisms. *Earth and Planetary Science Letters* 240, 642-655.
- Bouysse, P., Westercamp, D., 1990. Subduction of Atlantic aseismic ridges and Late Cenozoic evolution of the Lesser Antilles island arc. *Tectonophysics* 175, 349-380.
- Canales, J.P., Tucholke, B.E., Xu, M., Collins, J.A., DuBois, D.L., 2008. Seismic evidence for large-scale compositional heterogeneity of oceanic core complexes. *Geochemistry, Geophysics, Geosystems* 9, Q08002.
- Eccles, J.D., White, R.S., Christie, P.A.F., 2009. Identification and inversion of converted shear waves: case studies from the European North Atlantic continental margins. *Geophysical Journal International* 179, 381-400.
- Klein, F., Grozeva, N.G., Seewald, J.S., McCollom, T.M., Humphris, S.E., Moskowitz, B., Berquo, T.S., Kahl, W.A., 2015. Experimental constraints on fluid-rock reactions during incipient serpentinization of harzburgite. *American Mineralogist* 100, 991-1002.
- Li, L., Collier, J., Henstock, T., Goes, S., 2024. Downward continued ocean bottom seismometer data show continued hydrothermal evolution of mature oceanic upper crust. *Geology* 52, 717-722.
- Liu, M., Gerya, T., Rozel, A., 2025. The Effect of Brittle-Ductile Weakening on the Formation of Faulting Patterns at Mid-Ocean Ridges. *Tectonics* 44.
- Liu, M., Gerya, T., Rozel, A.B., 2022. Self-organization of magma supply controls crustal thickness variation and tectonic pattern along melt-poor mid-ocean ridges. *Earth and Planetary Science Letters* 584.
- Merdith, A.S., Atkins, S.E., Tetley, M.G., 2019. Tectonic Controls on Carbon and Serpentinite Storage in Subducted Upper Oceanic Lithosphere for the Past 320 Ma. *Front Earth Sc-Switz* 7, 332.

- Mevel, C., 2003. Serpentinization of abyssal peridotites at mid-ocean ridges. *Comptes Rendus Geoscience* 335, 825-852.
- Mukerji, T., Berryman, J., Mavko, G., Berge, P., 1995. Differential effective medium modeling of rock elastic moduli with critical porosity constraints. *Geophys Res Lett* 22, 555-558.
- Norris, A.N., 1985. A differential scheme for the effective moduli of composites. *Mechanics of Materials* 4, 1-16.
- Pichot, T., Patriat, M., Westbrook, G.K., Nalpas, T., Gutscher, M.A., Roest, W.R., Deville, E., Moulin, M., Aslanian, D., Rabineau, M., 2012. The Cenozoic tectonostratigraphic evolution of the Barracuda Ridge and Tiburon Rise, at the western end of the North America-South America plate boundary zone. *Marine Geology* 303, 154-171.
- Roest, W.R., Collette, B.J., 1986. The Fifteen Twenty Fracture Zone and the North American–South American plate boundary. *Journal of the Geological Society* 143, 833-843.
- Rüpke, L., Gaillard, F., 2024. The Geological History of Water: From Earth's Accretion to the Modern Deep Water Cycle. *Elements* 20, 253-258.
- Stein, S., Engeln, J.F., Wiens, D.A., Fujita, K., Speed, R.C., 1982. Subduction seismicity and tectonics in the Lesser Antilles Arc. *Journal of Geophysical Research* 87, 8642-8664.
- Taylor, M.A.J., Singh, S.C., 2002. Composition and microstructure of magma bodies from effective medium theory. *Geophysical Journal International* 149, 15-21.

Response to Reviewers' comments on Nature Communications manuscript NCOMMS-24-69268A

“Estimating excess bound water content due to serpentinisation in mature slow-spread oceanic crust using Vp/Vs” by Li et al.

Below reviewers' comments are in *blue italics* and our response is in black.

Reviewer #1 (Remarks to the Author):

The authors have undertaken a thorough revision of the manuscript based on the reviewer comments. All of my questions and concerns have been addressed, and it appears that the authors have included additional calculations to address one of the central comments of the other reviewer, which have also improved the manuscript.

As described in my previous review, I think this manuscript is very well written and illustrated, and it presents an important new result with implications for processes at slow-spreading mid-ocean ridges and subduction zones. I recommend publication.

Reviewer #2 (Remarks to the Author):

This is my second review of the Li et al. manuscript. The authors have thoughtfully addressed all my comments and I look forward to seeing the work published in Nature Communications.

We sincerely thank the reviewers for their insightful comments, which we believe have significantly improved the clarity and robustness of the manuscript. The positive assessments from both reviewers are encouraging, and we are delighted that they recognise the importance of our findings for processes at slow-spreading mid-ocean ridges and subduction zones. Their expertise and time invested in evaluating our manuscript have been invaluable, and we deeply appreciate their support for its publication in Nature Communications.